# xKV: Cross-Layer KV-Cache Compression via Aligned Singular Vector Extraction

## Abstract

Large Language Models (LLMs) with long context windows enable powerful applications but come at the cost of high memory consumption to store the key and value states (KV-Cache). Recent studies attempted to merge KV-Caches from multiple layers into shared representations, yet these approaches either require expensive pretraining or rely on per-token cosine similarity across layers, which may not always be observed in practice. We find that the dominant singular vectors are remarkably well-aligned across multiple layers of the KV-Cache. Exploiting this insight, we propose xKV, a post-training compression method that applies Singular Value Decomposition (SVD) on the KV-Cache of grouped layers. xKV consolidates the KV-Cache of multiple layers into a shared low-rank subspace, significantly reducing KV-Cache sizes. Through extensive evaluations on the RULER long-context benchmark with widely-used LLMs (*e.g.*, Llama-3.1 and Qwen2.5), xKV achieves up to $8\times$ KV-Cache compression rate while keeping the accuracy gap within 2–3 percentage points of the non-compressed baseline over a set of representative long-context tasks, and remains robust in multi-turn settings. Coupled with the designed *Selective Reconstruction* (SR) at decode time, xK-SR (keys only, values offloaded to CPU memory) yields 2.53% higher accuracy than the state-of-the-art system that combined token selection with single-layer SVD and delivers up to **3.23**$\times$ end-to-end generation speedups over full attention on an A100 GPU. At a similar accuracy level, xKV-SR (keys and values on GPU) achieves up to **4.23**$\times$ faster speedups. These results highlight xKV as a versatile, plug-and-play solution to alleviate both memory and latency bottlenecks in long-context LLM inference.

## 1 Introduction

Large language models (LLMs) (Touvron et al., 2023; OpenAI et al., 2024; Team et al., 2024; lla, 2024; Jiang et al., 2023; Anthropic, 2023) have revolutionized numerous artificial intelligence (AI) applications with advanced cognitive capabilities that were previously unattainable with conventional machine learning (ML) models. Recent efforts to extend the context lengths of LLMs have further expanded their potential: open-sourced models now support up to 1M tokens (Pekelis et al., 2024; Yang et al., 2025), and proprietary ones like Gemini push this limit even further to 10M tokens (Team et al., 2024). These extended context windows unlock a wide range of previously impractical applications, such as large-scale information retrieval and debugging or extending a large-scale codebase (DeepSeek-AI et al., 2025; Dubey et al., 2024; Yang et al., 2025; OpenAI et al., 2024).

However, this expanded capability on long-context introduces significant challenges, particularly in the management of key-value (KV) caches during inference (Fu, 2024; Li et al., 2024a). Typically, KV states are cached to avoid redundant computations; yet, under extended context lengths, the memory consumption of KV-Cache rapidly becomes prohibitive. This inflated memory footprint severely limits the number of concurrent inference requests, causing substantial throughput reduction. To address this, researchers have proposed various approaches to mitigate the large memory footprint of KV-Caches. These include quantization (Hooper et al., 2024; Liu et al., 2024c; Chen et al., 2025; Zhao et al., 2023), token eviction (Adnan et al., 2024; Ge et al., 2024; Xiao et al., 2024; Zhang et al., 2024b; Li et al., 2024b; Cai et al., 2024), and low-rank decomposition (Sun et al., 2024a; Chang et al., 2025; Zhang et al., 2024a; Yuan et al., 2023). These approaches have primarily focused on intra-layer redundancies that compress the KV-Cache of each layer separately. While this often yields

respectable per-layer compression, these methods do not utilize potential redundancy across layers (Gromov et al., 2024).

To exploit cross-layer redundancy, two main lines of work have emerged. The first, represented by Cross-Layer Attention (CLA) (Brandon et al., 2024) and YOCO (Sun et al., 2024b), introduces new architectures that share a single set of KV-Cache across groups of adjacent layers. While effective, these methods require architectural modifications and thus expensive pretraining from scratch, limiting their applicability to existing pretrained models. A second direction, exemplified by MiniCache (Liu et al., 2024b), operates in a post-hoc manner by merging adjacent layers' KV-Cache under the assumption of high cosine similarity, implemented via spherical linear interpolation (SLERP) (Shoemake, 1985). Our analysis, however, shows that such similarity, though present to some extent, is not consistently strong enough across layers to support robust compression, leading to nontrivial accuracy degradation in practice (see §3.1). Together, prior methods are limited either by costly pretraining or by fragile similarity assumptions, motivating the need for a new approach.

We revisit inter-layer similarity using Centered Kernel Alignment (CKA) (Kornblith et al., 2019). Our analysis reveals that, although the KV-Cache of adjacent layers exhibit low cosine similarity, their dominant singular vectors remain highly aligned (see §3.2). This observation enables us to share basis vectors across multiple adjacent layers' KV-Cache, yielding a more compact representation.

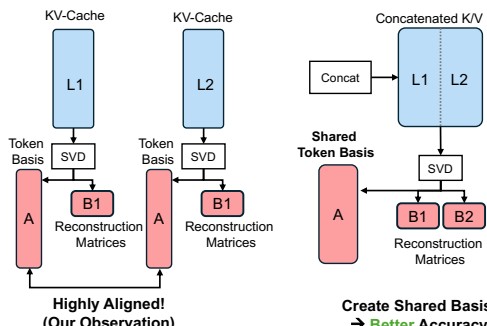

Building on this insight, we propose xKV, a fully *plug-and-play* compression method that requires no additional fine-tuning or architectural modifications. xKV simultaneously compresses the KV-Cache of multiple layers by extracting a *shared* set of singular vectors through cross-layer SVD, producing a compact token basis reused across adjacent layers as illustrate in Figure 1. To further reduce overhead at inference,

Figure 1: The token basis (singular vectors) of two different layers' KV-Cache are highly aligned. xKV concatenates adjacent layers and performs one SVD to obtain a shared basis $\mathbf{A}$ with layer-specific $\mathbf{B}_1$ and $\mathbf{B}_2$, improving accuracy at a fixed rank and reducing memory.

we introduce *Selective Reconstruction (SR)*: instead of reconstructing all tokens, we selectively reconstruct only those relevant to the query (§ 4.3). The pairing of cross-layer compression with SR substantially lowers reconstruction cost while preserving model accuracy, making xKV practical for real-world deployment.

To adapt xKV to diverse deployment requirements, we further design two decoding modes (§ 4.4). When the target application is latency-sensitive, we use xKV-SR, which compresses *both* keys and values and keeps them fully in GPU memory, yielding fastest decoding. When accuracy must be preserved, we use xK-SR, which compresses *keys only* while offloading values to CPU memory, delivering near-lossless accuracy with reduced GPU memory usage.

Extensive experiments on RULER with Llama (lla, 2024) and Qwen (Yang et al., 2024; 2025) models show that xKV achieves up to $8\times$ compression rate with minimal accuracy degradation (<3%), significantly outperforming representative token eviction and quantization baselines. With SR enabled, xK-SR yields **>2.5 percentage points** higher accuracy than state-of-the-art single-layer SVD systems. Most importantly, by keeping the compressed cache entirely on-device, xKV-SR eliminates PCIe bottlenecks, translating these efficiency gains into **3.6×** faster attention operation and up to **4.23×** higher end-to-end generation throughput over Full KV-Cache baseline with FlashAttention-2 CUDA kernel on Llama-3.1-8B.

## 2   RELATED WORK

**Low-Rank KV-Cache Compression.**   A broad line of research exploits the *low-rank nature* of the KV-Cache to reduce its memory footprint. For instance, Multi-Head Latent Attention (MLA) (Liu et al., 2024a; DeepSeek-AI et al., 2025) projects tokens onto a low-rank subspace and caches those latent representations instead of the original key and value states, however, MLA requires training the model from scratch. In contrast, several *post-training* techniques decompose the key/value parameter

matrices to obtain low-rank projection modules similar to MLA, such as ASVD (Yuan et al., 2023), Palu (Chang et al., 2025), and LoRC (Zhang et al., 2024a). Other methods decompose the KV-Cache directly: EigenAttention (Saxena et al., 2024) applies SVD to a calibration dataset to derive projection matrices, whereas ShadowKV (Sun et al., 2024a) performs online SVD to capture the dynamics of different contexts. In xKV, we also exploit the low-rank nature of KV-Cache. However, unlike prior methods focusing on per-layer compression, xKV further considers the shared information among multiple layers and extends the usage of low-rank projections to a new cross-layer dimension.

**Cross-Layer KV-Cache Optimization.**    Going beyond the intra-layer perspective, another stream of research explores inter-layer redundancy of KV-Cache (Brandon et al., 2024; Sun et al., 2024b; Wu & Tu, 2024; Liu et al., 2024b; Dong et al., 2025). CLA (Brandon et al., 2024) and YOCO(Sun et al., 2024b) both modify the Transformer model architecture so that later layers can directly reuse or reference KV states from earlier layers. LCKV (Wu & Tu, 2024) restricts full KV storage to a small subset of layers, foregoing caches in other layers. However, these methods rely on retraining or model fine-tuning, which makes them less flexible. Minicache (Liu et al., 2024b), in contrast, provides a flexible post-training alternative by merging the key and value tokens from adjacent similar layers using spherical linear interpolation. Our approach goes further by extracting shared singular vectors of multiple layers' KV-Caches, thereby enabling higher compression.

**Dynamic Token Selection and KV Offloading.**    A complementary line of work accelerates decoding by selecting a small subset of context tokens per step (dynamic sparse attention). Quest (Tang et al., 2024) proposes query-aware page selection to reduce attention cost without compressing the KV-Cache. ShadowKV (Sun et al., 2024a) stores a low-rank key cache on GPU, offloads values to CPU, and employs an accurate landmark-guided selector with a small static outlier set to reconstruct minimal sparse KV pairs on-the-fly, improving throughput under long contexts. In contrast, xKV targets cross-layer KV compression: we extract a shared low-rank token basis across adjacent layers and pair it with selective reconstruction. This lets us (i) match ShadowKV's "keys-only + offloaded values" regime via xK-SR, and (ii) run xKV-SR with both keys and values compressed on GPU, avoiding host–device transfer. Empirically, at matched token budgets, xK-SR/xKV-SR achieve higher accuracy than Quest and ShadowKV while offering stronger speedups when values remain on-device.

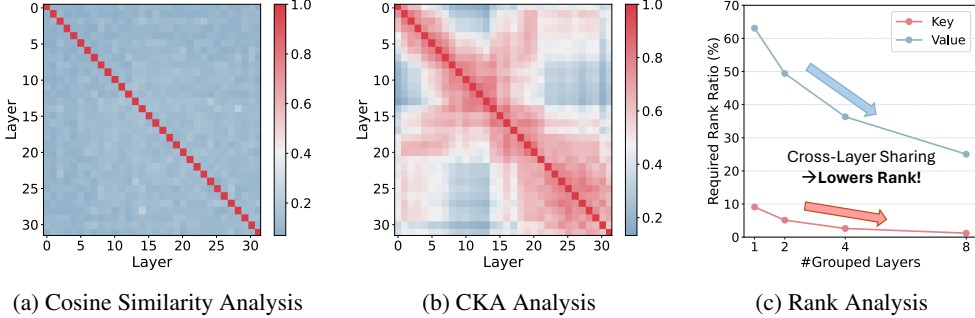

(a) Cosine Similarity Analysis          (b) CKA Analysis          (c) Rank Analysis

Figure 2: **(a)** Average Token-wise Cosine Similarity for value-caches across different layers. For each pair of layers, we compute the token-level cosine similarities between their embeddings and average these values into a single similarity score. **(b)** CKA Matrix for the value-cache. The higher (warmer) values indicate stronger singular vector alignment across layers. **(c)** Required rank ratio (percentage of total dimension) for capturing 95% of the cumulative eigenvalues in the key (red) and value (blue) matrices, plotted against the number of grouped layers. For each group, we horizontally concatenate the key/value caches and compute the rank needed to achieve 95% of the cumulative eigenvalues. As the grouping increases, a smaller rank (relative to total dimension) is required, implying a higher compression rate for the same level of information preservation. We perform these analyses on the KV-Cache obtained from Llama-3.1-8B-Instruct, using the multi-valued NIAH dataset from the RULER (Hsieh et al., 2024) benchmark.

## 3    ANALYSIS AND MOTIVATION

In this section, we examine the cross-layer similarity of KV-Caches with different metrics to reveal the motivation behind the design of xKV.

## 3.1 Cross-Layer Cosine Similarity (Prior Work)

To understand the assumption used in the previous work (Liu et al., 2024b), we first measure token-wise cosine similarity across various layer-pairs. The measurement on the cosine similarity is presented in Figure 2a. Notably, the adjacent layers exhibit low token-wise similarity. This modest similarity fundamentally limits the compression rate achieved by prior representative methods (Liu et al., 2024b).

## 3.2 Revisit Cross-Layer Similarity with CKA

While token-wise (cosine) similarity offers a local perspective, a more holistic view can reveal deeper patterns in how an entire KV-Cache is aligned across layers. Specifically, we adopt Centered Kernel Alignment (CKA) (Kornblith et al., 2019) to measure the similarity in the overall structure of two layers' KV-Caches. Concretely, for a layer $\ell$ with KV-Cache $\mathbf{X}_\ell \in \mathbb{R}^{n \times d}$, we first define the centered Gram matrix

$$\mathbf{G}_\ell = \mathbf{H}\,\mathbf{X}_\ell\,\mathbf{X}_\ell^\top\,\mathbf{H}, \quad \text{where} \quad \mathbf{H} = \mathbf{I}_n - \frac{1}{n}\,\mathbf{1}\,\mathbf{1}^\top.$$

Then, the *CKA* between two layers $\ell_1$ and $\ell_2$ is

$$\mathrm{CKA}\big(\mathbf{X}_{\ell_1}, \mathbf{X}_{\ell_2}\big) = \frac{\mathrm{trace}\big(\mathbf{G}_{\ell_1}\mathbf{G}_{\ell_2}\big)}{\sqrt{\mathrm{trace}\big(\mathbf{G}_{\ell_1}^2\big)\mathrm{trace}\big(\mathbf{G}_{\ell_2}^2\big)}}.$$

Unlike the token-wise cosine similarity metric, which simply compares corresponding token embeddings, CKA reflects the similarity of *the entire distribution* of token embeddings in each layer. If $\mathrm{CKA}(\mathbf{X}_{\ell_1}, \mathbf{X}_{\ell_2})$ is high, then the dominant left singular vectors of $\mathbf{X}_{\ell_1}$ are highly aligned to those of layer $\ell_2$ (*ref.* Appendix A). In other words, the basis vectors that define how the token varies in these two layers might be similar.

**Observation 1: Highly Aligned Basis.** In Figure 2b, we show the CKA value between each layers' KV-Cache of Llama-3.1-8B-Instruct. As shown in Figure 2b, many pairs of layers exhibit remarkably high CKA (red blocks) even though their token-wise cosine similarities are quite modest. This observation suggests that, although individual token embeddings differ across layers, the dominant singular vectors (*i.e., basis*) that span the KV-Cache are, in fact, *well-aligned*. Thus, focusing solely on the cosine similarity between pairs of token embeddings can underestimate the potential for *cross-layer* merging and compression.

## 3.3 Eigenvalue Analysis of KV-Cache

**Observation 2: Horizontally Concatenated KV-Caches Exhibit Lower Rank.** Motivated by the observation that different layers' basis are well aligned, we examine the rank to achieve a certain level of information preservation after horizontally concatenating the KV-Caches across multiple layers. Because each layer shows substantial cross-layer overlap (§3.2), a *single* set of low-rank basis vectors can effectively approximate the KV-Caches of all layers in the group. As illustrated in Figure 2c, a larger group size reduces the fraction of total rank needed to preserve the same cumulative eigenvalues. Compared with creating separate low-rank subspaces for each layer, this shared approach avoids storing nearly identical basis vectors multiple times, yielding a more compact yet expressive representation. In §4, we leverage these observations to propose our xKV method that achieves significantly higher compression ratios while preserving model accuracy.

## 4 Methodology: xKV

### 4.1 Notation

We consider a Transformer with $N$ decoder blocks and a long prompt of length $L$. Let $d$ denote the KV hidden size. Under GQA, $d = H_{\mathrm{kv}} \cdot d_h$ with $H_{\mathrm{kv}}$ KV heads and per-head width $d_h$. Because the same decomposition/reconstruction pipeline applies to both keys and values, we use a *unified* symbol

$$\mathbf{X}_\ell^\tau \in \mathbb{R}^{L \times d}, \qquad \tau \in \{K^{\mathrm{pre}}, V\},$$

to denote the cache of type $\tau$ at layer $\ell$. For RoPE models, we always decompose *pre-RoPE keys* ($\tau = K^{\mathrm{pre}}$) and re-apply RoPE after reconstruction.

**Decode-time head mapping and row selection.** Let $H_q$ be the number of query heads and $\rho : [H_q] \to [H_{\mathrm{kv}}]$ the GQA mapping from query heads to KV heads. At decode step $t$, for each layer $\ell$ and KV head $g$, we will use an index set $\mathcal{S}_{t,\ell,g} \subseteq [L]$ of selected prompt rows with $M_{t,\ell,g} = |\mathcal{S}_{t,\ell,g}|$ (§ 4.3).

## 4.2 Core Method: Cross-Layer SVD

Motivated by our empirical finding that the dominant left singular vectors of KV-Caches are well-aligned across adjacent layers (§ 3), we group layers into contiguous strides of size $G$:

$$\mathcal{G}_k = \{kG, \dots, kG + G - 1\}, \qquad k = 0, 1, \dots, \tfrac{N}{G} - 1.$$

For a group $\mathcal{G}_k = \{\ell_1, \dots, \ell_G\}$ and type $\tau \in \{K^{\mathrm{pre}}, V\}$, we horizontally concatenate the group's caches and compute a single low-rank factorization:

$$\left[ \mathbf{X}^\tau_{\ell_1}, \dots, \mathbf{X}^\tau_{\ell_G} \right] \approx \mathbf{U}^\tau_{k,r_\tau} \mathbf{S}^\tau_{k,r_\tau} \left( \mathbf{V}^\tau_{k,r_\tau} \right)^\top = \underbrace{\mathbf{A}^\tau_k}_{\in \mathbb{R}^{L \times r_\tau}} \left[ \underbrace{\mathbf{B}^\tau_{\ell_1}}_{\in \mathbb{R}^{r_\tau \times d}} \cdots \underbrace{\mathbf{B}^\tau_{\ell_G}}_{\in \mathbb{R}^{r_\tau \times d}} \right], \qquad (1)$$

where $\mathbf{A}^\tau_k = \mathbf{U}^\tau_{k,r_\tau} \mathbf{S}^\tau_{k,r_\tau}$ is the *shared token basis* for the group, and $\mathbf{B}^\tau_\ell$ are layer-specific reconstruction matrices. Compared to single-layer SVD, this *cross-layer* factorization learns a shared subspace across adjacent layers and is effective for *both keys and values*. Each layer-specific reconstruction matrix can also be view as the concatenation of KV-head specific reconstruction matrices:

$$\mathbf{B}^\tau_\ell = \left[ \mathbf{B}^\tau_{\ell,1} \cdots \mathbf{B}^\tau_{\ell,H_{\mathrm{kv}}} \right], \qquad \mathbf{B}^\tau_{\ell,g} \in \mathbb{R}^{r_\tau \times d_h}$$

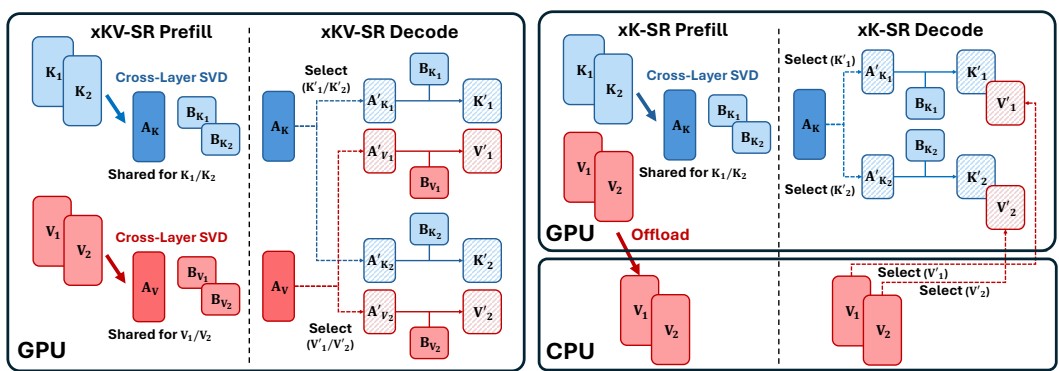

Figure 3: Illustration of different optimized operation modes. `xKV-SR` design (**Left**) keeps both low-rank key and value caches on the GPU. `xK-SR` design (**Right**) keeps the low-rank key cache on the GPU and offloads the full value cache to the CPU.

## 4.3 Process During Inference

**Prefill Compression.** During prefill, we compute (1) *separately* for $\tau = K^{\mathrm{pre}}$ and $\tau = V$ for every group $k$:

$$\left\{ \mathbf{A}^K_k, \{ \mathbf{B}^K_\ell \}_{\ell \in \mathcal{G}_k} \right\}, \qquad \left\{ \mathbf{A}^V_k, \{ \mathbf{B}^V_\ell \}_{\ell \in \mathcal{G}_k} \right\}.$$

We perform the decomposition online during prefill to capture prompt dynamics (the added cost is a small fraction of prefill and diminishes as $L$ grows). Empirically, the online cross-layer SVD accounts for only 3.9% of prefill time at sequence length of 128K (See Appendix C.1). Newly generated tokens are left uncompressed by default (their length is typically $\ll L$ in long-context use); for very long generations, we may reapply cross-layer SVD to those tokens.

**Dense reconstruction (baseline cost).** A direct use of the factors would, for each $\ell \in \mathcal{G}_k$ and head $g$, reconstruct all $L$ rows:

$$\widehat{\mathbf{X}}^\tau_{\ell,g} = \mathbf{A}^\tau_k \mathbf{B}^\tau_{\ell,g}.$$

For keys in RoPE models, we then set $\widehat{\mathbf{K}}^{\mathrm{rope}}_{\ell,g} = \mathrm{RoPE}\left( \widehat{\mathbf{X}}^{K^{\mathrm{pre}}}_{\ell,g} \right)$ by applying RoPE per row using its original position index. This dense strategy reconstruction FLOPs $\mathbf{AB}$ cost that scales with sequence $L$ at every step (Appendix D.5). Despite the memory saving that the decomposition can offer, this additional computation cost can pose an extra latency overhead during decoding.

**Selective reconstruction.** Prior work shows that LLMs exhibit strong *attention sparsity* during decoding, with most queries attending only to a small subset of context tokens (Sun et al., 2024a; Tang et al., 2024; Cai et al., 2024). Inspired by this characteristic, we leverage this inherent sparsity nature and reconstruct only the tokens that are likely to matter at that step. Specifically, at step $t$ we reconstruct only rows in $\mathcal{S}_{t,\ell,g} \subseteq [L]$:

$$\widehat{\mathbf{X}}_{\ell,g}^{\tau}[\mathcal{S}_{t,\ell,g}, :] = \mathbf{A}_k^{\tau}[\mathcal{S}_{t,\ell,g}, :] \, \mathbf{B}_{\ell,g}^{\tau}. \tag{2}$$

For any query head $h$ with $\rho(h) = g$, attention is then computed using $\widehat{\mathbf{X}}_{\ell,g}^{\tau}$ restricted to $\mathcal{S}_{t,\ell,g}$. (For RoPE models, we decompose pre-RoPE keys and apply RoPE after reconstruction.) In our implementation, the sets $\mathcal{S}_{t,\ell,g}$ are produced by a landmark-guided Top-$k$ chunk selector with a small static outlier set (Sun et al., 2024a). We provide the detailed workflow on how the indices $\mathcal{S}_{t,\ell,g}$ in Appendix B.1 and further analysis on the reconstruction FLOPs in Appendix D.5.

## 4.4 OPERATION MODES

We design two operation modes, `xKV-SR` and `xK-SR`, optimized for different scenarios. The overview is presented in Figure 3.

**Joint key–value compression with selective reconstruction (`xKV-SR`).** Leveraging cross-layer SVD, `xKV` can effectively compress *both* keys and values while maintaining strong accuracy, reducing the total KV footprint in device memory. With effective compression, we can fit the entire compressed KV on GPU's memory and avoid the necessity of KV-Cache offloading that induces host-device transfer, which is crucial when host–device bandwidth is limited (e.g., PCIe-only servers) or on unified-memory/embedded platforms (e.g., Jetson-class devices), allowing more requests per GPU and lower end-to-end latency.

**Key-only compression with selective reconstruction and value offloading (`xK-SR`).** When host-to-device bandwidth is sufficient (*e.g.*, 900GB/s on GB200 Goldwasser et al. (2024)), we adopt a key-only compression strategy that offloads the value cache to CPU memory, similar to ShadowKV (Sun et al., 2024a). Our analysis (Figure 2c) shows that values are relatively high-rank and more sensitive to compression, so leaving them uncompressed preserves accuracy. To mitigate the added memory cost of this design, we overlap key reconstruction (Eq. 2) with host-device value transfers, effectively hiding reconstruction latency behind data movement. Unlike ShadowKV, however, `xK-SR` leverages `xKV`'s *cross-layer* key factorization, yielding higher accuracy under the same memory budget.

## 5 ACCURACY EVALUATIONS

**Models.** We evaluate `xKV` on three widely used language models using Grouped-Query Attention (GQA): Llama-3.1-8B-Instruct (Dubey et al., 2024) (8 KV heads) and Qwen2.5-7B-Instruct-1M (Yang et al., 2025) (4 KV heads). In Appendix E, we also evaluate `xKV` on DeepSeek-Coder-V2-Lite-Instruct (Dai et al., 2024) with Multi-head Latent Attention (MLA) and Mixture-of-Experts (MoE) to demonstrate `xKV`'s high compatibility with emerging efficient Transformer architectures.

**Datasets.** We select RULER (Hsieh et al., 2024) as our major benchmark, which features complex tasks such as retrieval, multi-hop tracking, and question-answering. We also evaluate our approach using Needle In A Haystack (NIAH) (Kamradt, 2023) under multi-turn setups. We also provide the LongBench evaluation in the Appendix D.2.

**Baselines.** We compare `xKV` with the baselines in two scenarios. Firstly, the pure KV-Compression without selective reconstruction for reducing KV-Cache memory footprint. In this scenario, we compare against six baselines: (1) MiniCache (Liu et al., 2024b), the inter-layer compression method based on cosine similarity cross-layer. (2) Single SVD (Sun et al., 2024a), which compresses KV-Cache by factorizing each layer's key and value caches independently without exploiting the cross-layer similarity. (3) Token eviction baselines PyramidKV (Cai et al., 2024) and SnapKV (Li et al., 2024b). (4) A 2-bit quantization method KIVI (Liu et al., 2024c). (5) A token selection methodology, StreamingLLM (Xiao et al., 2024), Quest (Tang et al., 2024), that entails dynamic

Table 1: KV-Cache Compression Results: Performance of different methods on the RULER benchmark evaluated at a context length of 64K. xKV consistently achieves a higher accuracy than the Full Attns at the same compression rate or even at a significantly higher compression rate.

| Method | Comp. | N-S1 | N-S2 | N-MK1 | N-MK2 | N-MQ | N-MV | QA-1 | QA-2 | VT | FWE | Avg. |
|---|---|---|---|---|---|---|---|---|---|---|---|---|
| **Llama-3.1-8B-Instruct** | | | | | | | | | | | | |
| Full Attn | 1.00 | 100.00 | 100.00 | 98.96 | 97.92 | 98.96 | 97.66 | 83.33 | 59.38 | 97.29 | 85.42 | 91.89 |
| MiniCache | 1.30 | 89.58 | 66.67 | 43.75 | 10.42 | 14.06 | 21.35 | 61.46 | 35.42 | 49.38 | 58.33 | 45.04 |
| KIVI-2 | 7.10 | 100.00 | 96.88 | 98.96 | 90.63 | 91.41 | 89.58 | 80.21 | 55.21 | 81.46 | 84.38 | 86.87 |
| StreamingLLM | 8.00 | 15.63 | 12.50 | 13.54 | 13.54 | 14.58 | 17.97 | 56.25 | 45.83 | 9.58 | 94.10 | 29.35 |
| PyramidKV | 8.00 | 100.00 | 100.00 | 100.00 | 96.88 | 100.00 | 98.44 | 83.33 | 57.29 | 95.42 | 68.06 | 89.94 |
| SnapKV | 8.00 | 100.00 | 100.00 | 98.96 | 94.79 | 100.00 | 97.66 | 83.33 | 58.33 | 95.00 | 68.75 | 89.68 |
| Single SVD | 8.40 | 25.00 | 51.04 | 61.46 | 96.88 | 28.91 | 44.79 | 47.92 | 36.46 | 3.54 | 61.11 | 45.71 |
| xKV (Ours) | 8.03 | 100.00 | 96.88 | 97.92 | 97.92 | 96.09 | 96.62 | 78.13 | 56.25 | 86.67 | 78.47 | 88.50 |
| **Qwen2.5-7B-Instruct-1M** | | | | | | | | | | | | |
| Full Attn | 1.00 | 100.00 | 100.00 | 100.00 | 100.00 | 100.00 | 95.83 | 84.38 | 60.42 | 90.63 | 86.81 | 91.81 |
| MiniCache | 1.30 | 26.04 | 0.00 | 0.00 | 0.00 | 0.00 | 0.00 | 12.50 | 14.58 | 0.42 | 3.47 | 5.70 |
| KIVI-2 | 7.10 | 0.00 | 2.08 | 3.13 | 13.54 | 0.00 | 0.78 | 48.96 | 43.75 | 36.46 | 40.63 | 18.93 |
| StreamingLLM | 8.00 | 15.63 | 12.50 | 12.50 | 9.38 | 14.84 | 17.71 | 46.88 | 43.75 | 13.13 | 89.24 | 27.56 |
| PyramidKV | 8.00 | 100.00 | 93.75 | 96.88 | 16.67 | 90.37 | 80.73 | 84.38 | 59.38 | 89.17 | 76.39 | 78.77 |
| SnapKV | 8.00 | 100.00 | 96.88 | 97.92 | 31.25 | 95.31 | 83.07 | 84.38 | 59.38 | 91.25 | 80.56 | 82.00 |
| Single SVD | 8.40 | 100.00 | 97.92 | 96.88 | 98.96 | 97.40 | 91.15 | 64.58 | 56.25 | 73.75 | 61.46 | 83.84 |
| xKV (Ours) | 8.03 | 100.00 | 100.00 | 100.00 | 98.96 | 100.00 | 90.63 | 80.21 | 58.33 | 82.08 | 81.94 | 89.22 |

token selection. (6) A state-of-the-art baseline, ShadowKV (Sun et al., 2024a), that applies single-layer SVD compression on keys, offloads values to CPU memory, and performs token selection.

**Setup.** For xKV variants, we set the rank for key $r_{K^{pre}} = 384$ and $r_V = 576$ if value compression is applied. We use `torch.svd_lowrank` API from PyTorch for performing decomposition. We set the cross-layer group size to be 4 as the default setting. For baseline, we align MiniCache's official settings to merge half of the layers, from the middle to the end of the LLM, and vary the compression rate by adjusting the layer index at which merging begins. For the token eviction (e.g., SnapKV, PyramidKV) and quantization baseline (KIVI), we adopt the implementation from MInference (Jiang et al., 2024; Li et al., 2025) library. We keep the newly generated tokens uncompressed for all comparison targets to ensure fair comparison. Unless specified, we calculate the compression rate by assuming a context length of 64k.

## 5.1 RESULTS ON RULER DATASETS

**KV-Cache Compression Results.** Table 1 reports the performance of xKV and several representative compression methods on the RULER benchmark at a 64K context length. As shown in Table 1, MiniCache suffers dramatic accuracy loss even at a modest 1.3× compression rate. This degradation echos our finding in §3.1), the token-wise cosine similarity in KV-Cache across adjacent layers is generally low. Compared to single-layer SVD compression, xKV yields substantial accuracy gains: at an 8× compression rate, xKV improves average accuracy by 43% on Llama-3.1-8B-Instruct and by 8% on Qwen2.5-7B-Instruct-1M, demonstrating its superior information preservation by exploiting the inherent alignment of KV-Cache representations across layers.

In comparison with token-eviction methods, xKV achieves 88.50% accuracy on Llama-3.1-8B-Instruct at 8.03× compression, closely matching SnapKV. On Qwen2.5-7B-Instruct-1M, however, both SnapKV and Pyramid incur noticeable accuracy degradation. We attribute this to Qwen2.5's inherently more compact KV cache—due to its smaller number of KV heads—which makes information preservation more challenging. Despite this, xKV attains 89.22% average accuracy, narrowing the gap to the non-compressed baseline to just 2.6%. Moreover, xKV surpasses the quantization baseline KIVI-2 by 1.7% on Llama-3.1-8B while maintaining accuracy on Qwen2.5, where KIVI-2 suffers significant drops. Finally, as shown in Appendix D.4, our approach can be combined with quantization to further increase compression without sacrificing accuracy.

**Results on Multi-turn Conversation Datasets.** We test our method using a multi-turn Needle-In-A-Haystack (NIAH) benchmark and compare its efficacy against token eviction–based approaches (e.g., SnapKV and PyramidKV). We conduct the evaluation at context length of 64K. Figure 4 shows results on Llama-3.1-8B-Instruct. SnapKV and PyramidKV both suffer steep declines after the first turn because they evict tokens using the initial attention patterns of the first query and cannot recover

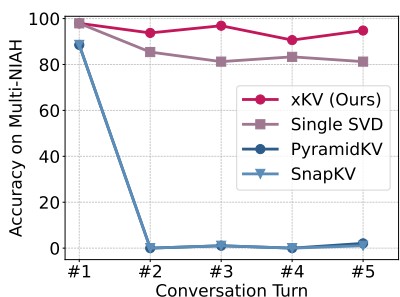

Figure 4: Accuracy of each conversation turn on Multi-turn NIAH. PyramidKV, SnapKV, and `xKV` are all at a compression rate of 8×.

Table 2: Accuracy across different group sizes on RULER with Llama-3.1-8B-Instruct. We align the rank setting with Table 1 and Table 3 for group size 4. For group sizes 1, 2, and 8, we scaled the rank linearly to maintain the same compression rate, with $(r_{K^{pre}}, r_V) = (96, 144)$ and $(192, 288)$, respectively.

| Group Size | xKV | xK-SR | xKV-SR |
|---|---|---|---|
| 1 | 45.71 | 87.17 | 72.27 |
| 2 | 75.15 | 88.43 | 86.06 |
| 4 | 88.50 | 89.70 | 89.69 |
| 8 | 88.91 | 89.74 | 89.72 |

Table 3: KV-Cache Compression with Selective Reconstruction Results: Accuracy of different methods on the RULER benchmark at a context length of 64K. Here, "Comp." indicates the total KV-Cache reduction, while the number in parentheses shows the effective GPU memory reduction considering KV-Cache offloading. ShadowKV* refers to a variant of ShadowKV that additionally compresses the value cache.

| Method | Comp. | N-S1 | N-S2 | N-MK1 | N-MK2 | N-MQ | N-MV | QA-1 | QA-2 | VT | FWE | Avg. |
|---|---|---|---|---|---|---|---|---|---|---|---|---|
| \multicolumn{13}{c}{Llama-3.1-8B-Instruct} | | | | | | | | | | | | |
| Full Attn | 1.00 | 100.00 | 100.00 | 98.96 | 97.92 | 98.96 | 97.66 | 83.33 | 59.38 | 97.29 | 85.42 | 91.89 |
| Quest | 1.00 (8.00) | 93.75 | 90.63 | 96.88 | 87.50 | 94.27 | 85.42 | 83.33 | 57.29 | 77.71 | 81.94 | 84.87 |
| ShadowKV | 1.64 (9.08) | 100.00 | 100.00 | 98.96 | 97.92 | 96.88 | 94.53 | 82.29 | 60.42 | 66.04 | 74.65 | 87.17 |
| xK-SR (Ours) | 1.63 (8.90) | 100.00 | 100.00 | 98.96 | 97.92 | 98.44 | 95.31 | 83.33 | 60.42 | 87.92 | 74.65 | 89.70 |
| ShadowKV* | 5.51 | 100.00 | 76.04 | 75.00 | 97.92 | 54.43 | 45.83 | 81.25 | 57.29 | 47.29 | 74.31 | 70.94 |
| xKV-SR (Ours) | 5.35 [1] | 100.00 | 100.00 | 98.96 | 97.92 | 98.44 | 95.57 | 82.29 | 60.42 | 87.29 | 76.04 | 89.69 |

context for later queries (Li et al., 2025). In contrast, our `xKV` maintains stable performance across all turns and consistently preserves critical information.

**KV-Cache Compression with Selective Reconstruction Results** In Table 3, we compare `xK-SR`, `xKV-SR`, and two representative token selection baselines, Quest and ShadowKV, using the RULER benchmark at a 64K context length for Llama-3.1-8B-Instruct. For a fair comparison, we fix the token budget (*i.e.*, the number of tokens selected for each decoding step) to be 2k for evaluation targets. Compared with Quest, both `xK-SR` and `xKV-SR` showcase superior accuracy with around 4% higher in average. As Quest does not entail KV-Cache compression but only dynamic loading, it does not reduce the size of the KV-Cache and necessitates KV-Cache offloading to avoid out-of-memory (OOM). Compared against ShadowKV, `xK-SR` extends its by replacing the single-layer SVD compression key cache with a cross-layer alternative. At a 1.64× KV-compression rate (8.9× GPU memory reduction considering value offloading), `xKV-SR` closes the accuracy gaps from 4.7% to around 2.1%, demonstrating xKV's better capability in preserving information. Leveraging the cross-layer alignment that we observed, `xKV-SR` is able to compress and reduce the KV-Cache to a significant 5.35× while maintaining 89.69% accuracy, roughly 19% higher than ShadowKV*. This enables retaining all tensors on GPUs and unlocking the faster inference that avoids the host-device transfer, which improves decoding efficiency over offloading scenarios (See Section 6).

**Impact of xKV on Compressing Value and Key Only.** To understand how `xKV` affects key and value compression, we conduct ablation experiments on four subtasks from RULER (Hsieh et al., 2024) to evaluate how `xKV` (cross-layer low-rank SVD) affects key and value compression. We show the results in Figure 5. Overall, `xKV` consistently boosts accuracy under varying compression rates. Also, keys exhibit higher compressibility than values, matching the eigenvalue analysis in Figure 2c. A closer inspection of the results reveals that the achievable compression ratio appears

---

[1] This set up have the 8× compressed KV-Cache using cross-layer SVD. The final compression rate is calculated, including the memory cost of the landmark for computing selective indices. See Appendix D.5 for more details.

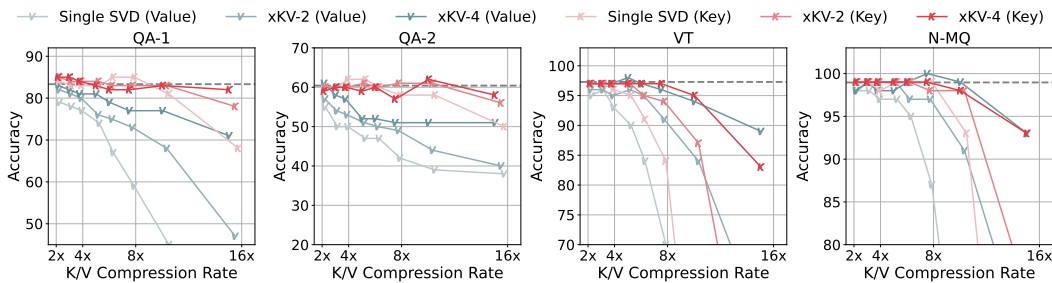

Figure 5: Accuracy comparison of applying different methods to key and value separately on Llama-3.1-8B-Instruct using RULER benchmark. The number after `xKV` denotes the cross-layer group size.

to be task-dependent. On the questions-answering subtasks (QA-1 and QA-2) `xKV` can push the compression rate to $16\times$ while still preserving performance. In Variable Tracking (VT) and NIAH multi-queries (N-MQ) (Kamradt, 2023), accuracy begins to decline beyond $8\times$ compression; however, in these same tasks, values tolerate compression more easily than in QA subtasks. These observations underscore how different tasks may demand different "sweet spots" for key versus value compression. In `xKV`, we employ a fixed compression ratio for all different tasks. Exploring task-specific or context-aware (Liu et al., 2023b; Akhauri et al., 2025; 2024) rank allocation is a promising avenue for future work.

**Impact of Cross-layer Group Size to Accuracy.**    To quantify the impact of cross-layer compression, we conduct a group size ablation on the RULER benchmark at a fixed compression rate (Table 2). For example, `xKV` improves from 45.71% with group size 1 to 75.15% at size 2, and further to 88.50% at size 4. Similar trends are observed for `xK-SR` and `xKV-SR`, where performance likewise climbs steadily as group size increases. These results confirm that sharing across more layers consistently enhances reconstruction fidelity under an identical compression rate. However, at a group size of 8, the accuracy of `xKV`, `xK-SR`, and `xKV-SR` all saturates, with accuracy nearly identical to that at a size of 4. Therefore, we use a group size of 4 in all main experiments.

# 6 EFFICIENCY STUDIES

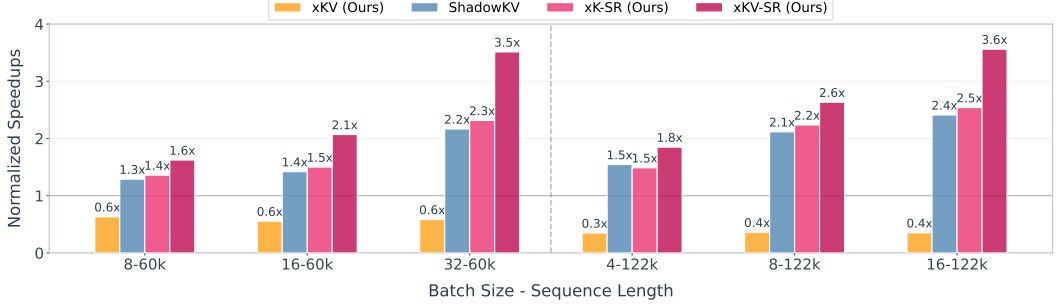

Figure 6: Attention latency evaluation. Normalized speedup relative to FlashAttention-2.

**Setup.**    We evaluate performance on Llama-3.1-8B (GQA) using an A100 (80GB). Figure 7 reports end-to-end generation throughput, while Figure 6 isolates the normalized attention latency relative to FlashAttention-2.

**Dense Reconstruction (`xKV`).**    `xKV` reduces memory usage and enables larger batch sizes than Full Attention, but its runtime is limited by the cost of reconstructing dense KV-Cache tensors. As Figure 6 shows, dense reconstruction cost grows with sequence length, which increases attention latency ($0.6\times$ speed at 64k and $0.3\times$ at 128k). This transition from memory-bound to compute-bound

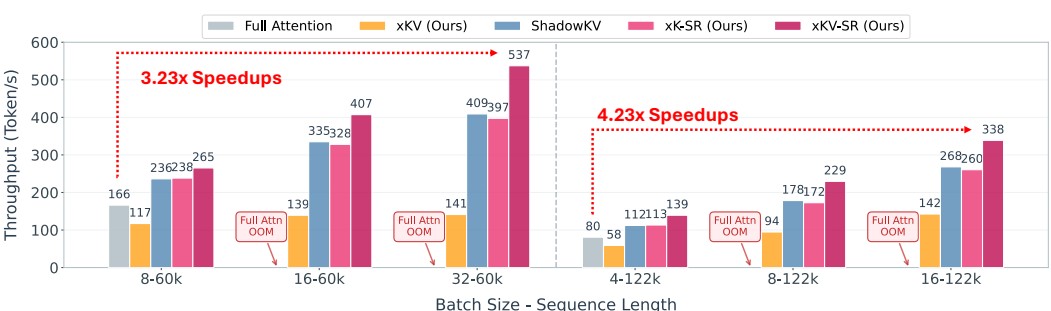

Figure 7: Generation throughput (tokens/s) on an A100.

execution is further reflected in the end-to-end throughput (Figure 7). When memory is no longer a bottleneck, `xKV` performs similarly to or slightly worse than the baseline.

**Selective Reconstruction (`xKV-SR`).** `xKV-SR` addresses the compute bottleneck via selective reconstruction, delivering the highest performance across all metrics. By keeping compressed KV-Cache entirely on the GPU, it achieves consistent attention latency gains, reaching up to $3.6\times$ speedup in Figure 6. This translates directly to generation throughput (Figure 7), where `xKV-SR` attains up to $3.23\times$ and $4.23\times$ speedups at 60k and 122k tokens, respectively.

**Selective reconstruction with offloaded values (`xK-SR` and ShadowKV).** Both `xK-SR` and ShadowKV operate in a "keys-only compression + offloaded values" regime. This regime is the setups with most memory saving. However, their performance tracks closely and is strictly bounded by PCIe bandwidth rather than compute. Comparisons in Figure 6 confirm that reconstructing compressed values on-chip (`xKV-SR`) is significantly faster than fetching uncompressed values over PCIe (`xK-SR` and ShadowKV), making it the superior choice for high-throughput, long-context decoding.

## 7 LIMITATIONS AND FUTURE WORK

**Long Generation Scenario.** Our study focuses on the long-prefill setting, where only the initial context is compressed while tokens generated during decoding remain uncompressed. This regime covers many long-context applications (e.g., information retrieval (Perplexity, 2025) and database QA), but it does not address test-time scaling under extended generation, which the cumulative KV-Cache can also become the bottleneck. We leave to future work how to leverage the observed cross-layer alignment of the KV-cache's dominant singular vectors and proposed cross-layer SVD to tackle long-generation scenarios.

## 8 CONCLUSION

We introduce `xKV`, a plug-and-play compression method for key-value (KV) caches that exploits inter-layer redundancy. Our approach reveals that KV-Caches across different layers share highly aligned basis vectors. Leveraging this property, we apply a cross-layer SVD to compress multiple KV-Caches into a shared low-rank subspace. Experiments demonstrate that `xKV` outperforms accuracy on all other compression methods, including representative inter-layer approaches and intra-layer methods such as quantization, token eviction, and single-layer SVD. At roughly 8× compression, `xKV` keeps average accuracy within 2–3 percentage points of the non-compressed baseline, and it remains robust in multi-turn settings. With *Selective Reconstruction* (SR), our fastest alternative `xKV-SR` reaches up to **4.23×** faster generation speed on A100 GPU, highlighting `xKV` as a practical approach to reduce both memory footprint and latency for long-context LLM inference.

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

# A CKA AND INDICATION OF ALIGNED LEFT SINGULAR VECTORS

## A.1 NOTATION AND DEFINITIONS

For each layer $\ell$, let
$$\mathbf{X}_\ell \in \mathbb{R}^{n \times d},$$
where each of the $n$ rows corresponds to a token (data point). Define the centering matrix
$$\mathbf{H} = \mathbf{I}_n - \frac{1}{n}\mathbf{1}\mathbf{1}^\top,$$
which subtracts the (row) mean from each token embedding. We define the *centered* embeddings
$$\widetilde{\mathbf{X}}_\ell = \mathbf{H}\mathbf{X}_\ell,$$
and the *centered Gram matrix*
$$\mathbf{G}_\ell = \widetilde{\mathbf{X}}_\ell\widetilde{\mathbf{X}}_\ell^\top \in \mathbb{R}^{n \times n}.$$
Because $\mathbf{G}_\ell$ is symmetric and positive semidefinite, its largest-eigenvalue directions capture the most "energetic" dimensions of $\widetilde{\mathbf{X}}_\ell$.

Given two layers $\ell_1$ and $\ell_2$, the *Centered Kernel Alignment (CKA)* between their token embeddings is
$$\mathrm{CKA}\big(\mathbf{X}_{\ell_1}, \mathbf{X}_{\ell_2}\big) = \frac{\mathrm{trace}\big(\mathbf{G}_{\ell_1}\mathbf{G}_{\ell_2}\big)}{\sqrt{\mathrm{trace}\big(\mathbf{G}_{\ell_1}^2\big)\,\mathrm{trace}\big(\mathbf{G}_{\ell_2}^2\big)}},$$
which measures how similarly $\mathbf{G}_{\ell_1}$ and $\mathbf{G}_{\ell_2}$ encode pairwise relationships (dot products) among the $n$ token embeddings.

## A.2 SVD PERSPECTIVE AND OVERLAP

**SVD of centered embeddings.** Consider the (compact) SVD of $\widetilde{\mathbf{X}}_\ell$:
$$\widetilde{\mathbf{X}}_\ell = \mathbf{U}_\ell\,\boldsymbol{\Sigma}_\ell\,\mathbf{V}_\ell^\top,$$
where:
$$\mathbf{U}_\ell \in \mathbb{R}^{n \times r}\quad(\text{orthonormal columns}),\quad \boldsymbol{\Sigma}_\ell = \mathrm{diag}(\sigma_1, \ldots, \sigma_r),\quad \mathbf{V}_\ell \in \mathbb{R}^{d \times r}\quad(\text{orthonormal columns}),$$
and $r \le \min(n, d)$ is the rank. Then the centered Gram matrix factors as
$$\mathbf{G}_\ell = \widetilde{\mathbf{X}}_\ell\widetilde{\mathbf{X}}_\ell^\top = \mathbf{U}_\ell\,\boldsymbol{\Sigma}_\ell^2\,\mathbf{U}_\ell^\top,$$
so the columns of $\mathbf{U}_\ell$ are exactly the eigenvectors of $\mathbf{G}_\ell$, and $\sigma_i^2$ are the corresponding eigenvalues.

**CKA in terms of left singular vectors.** Let $\widetilde{\mathbf{X}}_{\ell_1} = \mathbf{U}_{\ell_1}\,\boldsymbol{\Sigma}_{\ell_1}\,\mathbf{V}_{\ell_1}^\top$ and $\widetilde{\mathbf{X}}_{\ell_2} = \mathbf{U}_{\ell_2}\,\boldsymbol{\Sigma}_{\ell_2}\,\mathbf{V}_{\ell_2}^\top$. Then
$$\mathbf{G}_{\ell_1} = \mathbf{U}_{\ell_1}\,\boldsymbol{\Sigma}_{\ell_1}^2\,\mathbf{U}_{\ell_1}^\top,\quad \mathbf{G}_{\ell_2} = \mathbf{U}_{\ell_2}\,\boldsymbol{\Sigma}_{\ell_2}^2\,\mathbf{U}_{\ell_2}^\top.$$
We compute
$$\mathrm{trace}\big(\mathbf{G}_{\ell_1}\,\mathbf{G}_{\ell_2}\big) = \mathrm{trace}\Big(\mathbf{U}_{\ell_1}\,\boldsymbol{\Sigma}_{\ell_1}^2\,\mathbf{U}_{\ell_1}^\top\,\mathbf{U}_{\ell_2}\,\boldsymbol{\Sigma}_{\ell_2}^2\,\mathbf{U}_{\ell_2}^\top\Big) = \sum_{i=1}^{r_1}\sum_{j=1}^{r_2}\sigma_{\ell_1,i}^2\,\sigma_{\ell_2,j}^2\left(\mathbf{u}_{\ell_1}^{(i)\top}\mathbf{u}_{\ell_2}^{(j)}\right)^2,$$
where $\mathbf{u}_{\ell_1}^{(i)}$ and $\mathbf{u}_{\ell_2}^{(j)}$ are the $i$-th and $j$-th columns of $\mathbf{U}_{\ell_1}$ and $\mathbf{U}_{\ell_2}$, respectively. Meanwhile,
$$\mathrm{trace}\big(\mathbf{G}_{\ell_1}^2\big) = \sum_{i=1}^{r_1}\sigma_{\ell_1,i}^4,\qquad \mathrm{trace}\big(\mathbf{G}_{\ell_2}^2\big) = \sum_{j=1}^{r_2}\sigma_{\ell_2,j}^4.$$
Hence,
$$\mathrm{CKA}\big(\mathbf{X}_{\ell_1}, \mathbf{X}_{\ell_2}\big) = \frac{\sum_{i,j}\sigma_{\ell_1,i}^2\,\sigma_{\ell_2,j}^2\big(\mathbf{u}_{\ell_1}^{(i)\top}\mathbf{u}_{\ell_2}^{(j)}\big)^2}{\sqrt{\big(\sum_i\sigma_{\ell_1,i}^4\big)\big(\sum_j\sigma_{\ell_2,j}^4\big)}}.$$
Because the eigenvalues $\sigma_{\ell,i}^2$ reflect how "dominant" each left singular vector is, a **large CKA** value requires significant overlap $\big(\mathbf{u}_{\ell_1}^{(i)\top}\mathbf{u}_{\ell_2}^{(j)}\big)^2$ for the most important (largest-$\sigma^2$) directions, implying the principal subspaces of $\mathbf{G}_{\ell_1}$ and $\mathbf{G}_{\ell_2}$ align closely.

### A.3 CONCLUSION

In summary, when $\mathrm{CKA}(\mathbf{X}_{\ell_1}, \mathbf{X}_{\ell_2})$ is high, the dominant *left singular vectors* of $\widetilde{\mathbf{X}}_{\ell_1}$ and $\widetilde{\mathbf{X}}_{\ell_2}$ are well aligned. Since these vectors also serve as the *largest-eigenvalue* directions of the centered Gram matrices, high CKA implies that the *principal subspace* geometry of the token embeddings in layers $\ell_1$ and $\ell_2$ is *structurally* very similar—even if token-by-token (cosine) matches are small. Thus, CKA goes beyond individual token similarities, capturing **how** tokens vary collectively in a shared subspace.

## B IMPLEMENTATION DETAILS

### B.1 LANDMARK-GUIDED CHUNK SELECTOR FOR SELECTIVE RECONSTRUCTION

---

**Algorithm 1** Landmark Construction (Prefill)

---

**Require:** Post-RoPE keys $K_\ell^{\mathrm{rope}} \in \mathbb{R}^{H_{\mathrm{kv}} \times N \times d_h}$, chunk size $c$, optional #outliers $o$
**Ensure:** Landmarks $L_\ell \in \mathbb{R}^{H_{\mathrm{kv}} \times n_c \times d_h}$, optional outlier indices $\{\mathcal{O}_{\ell,g} \subseteq [n_c]\}_{g=1}^{H_{\mathrm{kv}}}$
 1: $n_c \leftarrow \lceil N/c \rceil$;
 2: **Chunking the sequence:** $\widetilde{K} \leftarrow \mathrm{View}(K_\ell^{\mathrm{rope}}) \in \mathbb{R}^{H_{\mathrm{kv}} \times n_c \times c \times d_h}$
 3: **Chunk means (landmarks):** $L_\ell \leftarrow \mathrm{mean}\left(\widetilde{K}, \mathrm{axis}=2\right) \in \mathbb{R}^{H_{\mathrm{kv}} \times n_c \times d_h}$
 4: **(Optional) Static outliers, per head:** $S_{\mathrm{cos}} \leftarrow \cos\left(\widetilde{K}, L_\ell \text{ broadcast along } c\right) \in \mathbb{R}^{H_{\mathrm{kv}} \times n_c \times c}$
 5: $\quad m \leftarrow \min(S_{\mathrm{cos}}, \mathrm{axis}=2) \in \mathbb{R}^{H_{\mathrm{kv}} \times n_c}$; $\quad I^{\mathrm{out}} \leftarrow \mathrm{ArgTopK}(-m, o)$; $\quad \mathcal{O}_{\ell,g} \leftarrow I^{\mathrm{out}}[g, :]$
 6: **return** $L_\ell$ and (optionally) $\{\mathcal{O}_{\ell,g}\}$

---

**Algorithm 2** Landmark-Guided Top-$k$ Chunk Selection (Decode)

---

**Require:** Landmarks $L_\ell \in \mathbb{R}^{H_{\mathrm{kv}} \times n_c \times d_h}$, queries $Q_{t,\ell} \in \mathbb{R}^{H_q \times d_h}$, GQA map $\rho : [H_q] \rightarrow [H_{\mathrm{kv}}]$, token budget $k$, chunk size $c$, optional outliers $\{\mathcal{O}_{\ell,g}\}$
**Ensure:** Per–KV head selected chunk indices $\{S_{t,\ell,g} \subseteq [n_c]\}_{g=1}^{H_{\mathrm{kv}}}$
 1: $k_{\mathrm{ch}} \leftarrow \lceil k/c \rceil$ $\qquad\qquad\qquad\qquad\qquad\qquad\qquad$ ▷ convert token budget to chunk budget
 2: **Scores to landmarks (batched MatMul):**
$$P \in \mathbb{R}^{H_q \times H_{\mathrm{kv}} \times n_c} \;\leftarrow\; \langle Q_{t,\ell}[:,\cdot],\, L_\ell[\cdot,:,\cdot] \rangle_{d_h} \big/ \sqrt{d_h}$$
 3: **Pool from query heads to KV heads (GQA):**
$$S[g,j] \;\leftarrow\; \max_{h:\,\rho(h)=g} P[h,g,j] \quad \text{for all } g \in [H_{\mathrm{kv}}],\ j \in [n_c]$$
 4: **Top-$k_{\mathrm{ch}}$ per KV head:** $\quad I \in \mathbb{R}^{H_{\mathrm{kv}} \times k_{\mathrm{ch}}} \;\leftarrow\; \mathrm{ArgTopK}(S, k_{\mathrm{ch}})$
 5: **(Optional) add static outliers:** $\quad S_{t,\ell,g} \;\leftarrow\; \mathrm{Union}\left(I[g,:], \mathcal{O}_{\ell,g}\right) \quad$ for each $g$
 6: **return** $\{S_{t,\ell,g}\}_{g=1}^{H_{\mathrm{kv}}}$

---

We adopted the landmark-guided selection techniques from ShadowKV (Sun et al., 2024a) to decide the token indices for selective reconstruction. We detail the workflow in the paragraphs below.

**Landmark construction (prefill).** At layer $\ell$, we split the post-RoPE key sequence into $n_c = \lceil N/c \rceil$ contiguous chunks of size $c$. For each KV head $g$ and chunk $j$, we define the *landmark* as the mean key of that chunk:
$$\ell_{j,g} \;=\; \frac{1}{|C_j|} \sum_{x \in C_j} K_{\ell,g}^{\mathrm{rope}}(x).$$

Optionally, we keep a tiny per-head *static outlier* set to guard against heterogeneous chunks whose mean is a weak representative. We identify these by computing the minimum within-chunk cosine similarity to the landmark,
$$r_{g,j} \;=\; \min_{x \in C_j} \cos\left(K_{\ell,g}^{\mathrm{rope}}(x),\, \ell_{j,g}\right),$$

and marking the $o$ chunks with the smallest $r_{g,j}$ as outliers for each head $g$. This metric indicates how well the landmark summarizes its chunk: lower values signal that at least one token is poorly captured by the mean, so those chunks are always considered during decoding. We summarize the procedure in Algorithm. 1.

**Landmark-guided selection (decode).** At each decode step $t$, given queries $Q_{t,\ell} \in \mathbb{R}^{H_q \times d_h}$, we score every chunk via a batched scaled dot-product between $Q_{t,\ell}$ and the landmarks. With grouped-query attention, scores are pooled from query heads to KV heads using the GQA map $\rho : [H_q] \to [H_{\mathrm{kv}}]$ by taking a max over the query heads mapped to each KV head. Given a token budget $k$ and chunk size $c$, we convert to a chunk budget $k_{\mathrm{ch}} = \lceil k/c \rceil$ and keep the top $k_{\mathrm{ch}}$ chunks per KV head. Optionally, we union these with the static outliers $\mathcal{O}_{\ell,g}$. The selected chunk indices are then expanded to row indices $\mathcal{S}_{t,\ell,g}$ and used to reconstruct only the corresponding tokens. We summarize the procedure in Algorithm. 2.

## C MORE LATENCY STUDIES

### C.1 ON-THE-FLY SVD OVERHEAD

Table 4 reports the latency of the prefilling phase as well as the cross-layer SVD on A6000 GPU. On a sequence length of $L = 64k$ tokens, the SVD accounts for 6.92% of the forward-pass time. This fraction steadily decreases as $L$ increases, reaching only 2.05% at $L = 256k$, where $L$ denotes the sequence length. The reduction can be attributed to the fact that the cost of attention grows quadratically with $L$, whereas the low-rank decomposition scales only linearly (Li et al., 2019). As a result, for very long contexts, the one-time decomposition performed during the prefill phase becomes practically negligible, contributing minimally to the overall computation time. Similar trends also hold on A100 GPU as demonstrated in Table 5.

Table 4: The latency data of on-the-fly SVD under different context lengths. Measured on an A6000 GPU with Qwen2.5-14B-Instruct. (Unit: seconds)

| Seqlen | 64k | 128k | 160k | 256k |
|---|---|---|---|---|
| Prefill Time | 39.02 | 122.30 | 182.54 | 425.42 |
| SVD time ($G$=2) | 1.98 (5.04%) | 3.48 (2.85%) | 4.37 (2.39%) | 6.36 (1.49%) |
| SVD time ($G$=4) | 2.70 (6.92%) | 4.76 (3.90%) | 5.89 (3.23%) | 8.74 (2.05%) |

Table 5: The latency data of on-the-fly SVD under different context lengths. Measured on an A100 GPU with Llama-3.1-8B. (Unit: seconds)

| Seqlen | 64k | 128k | 160k | 256k |
|---|---|---|---|---|
| Prefill Time | 18.98 | 47.87 | 71.55 | 159.27 |
| SVD time ($G$=2) | 1.90 (10.03%) | 3.48 (7.26%) | 4.38 (6.12%) | 6.35 (3.99%) |
| SVD time ($G$=4) | 2.56 (13.46%) | 4.75 (9.93%) | 5.89 (8.23%) | 8.74 (5.48%) |

## D MORE EXPERIMENTAL RESULTS

### D.1 MORE RESULTS ON RULER

**KV-Cache More Compression with Selective Reconstruction Results.** Table 6 reports results at different compression rates on the RULER benchmark. At the high compression setting (around $11.5\times$ effective GPU memory reduction), xK-SR outperforms ShadowKV by a striking 36%. This demonstrates that xKV-SR is significantly more effective at preserving performance under extreme compression.

Table 6: More KV-Cache Compression with Selective Reconstruction Results: Accuracy of different methods on the RULER benchmark at a context length of 64K. Here, "Comp." indicates the total KV-Cache reduction, while the number in parentheses shows the effective GPU memory reduction considering KV-Cache offloading. ShadowKV* refers to a variant of ShadowKV that additionally compresses the value cache.

| Method | Comp. | N-S1 | N-S2 | N-MK1 | N-MK2 | N-MQ | N-MV | QA-1 | QA-2 | VT | FWE | Avg. |
|---|---|---|---|---|---|---|---|---|---|---|---|---|
| **Llama-3.1-8B-Instruct** | | | | | | | | | | | | |
| Full Attn | 1.00 | 100.00 | 100.00 | 98.96 | 97.92 | 98.96 | 97.66 | 83.33 | 59.38 | 97.29 | 85.42 | 91.89 |
| Quest | 1.00 (8.00) | 93.75 | 90.63 | 96.88 | 87.50 | 94.27 | 85.42 | 83.33 | 57.29 | 77.71 | 81.94 | 84.87 |
| ShadowKV | 1.60 (7.94) | 100.00 | 100.00 | 100.00 | 97.92 | 99.22 | 95.83 | 83.33 | 59.38 | 78.33 | 73.96 | 88.80 |
| xK-SR (Ours) | 1.59 (7.76) | 100.00 | 100.00 | 98.96 | 97.92 | 98.70 | 96.35 | 82.29 | 61.46 | 88.33 | 75.69 | 89.97 |
| ShadowKV | 1.64 (9.08) | 100.00 | 100.00 | 98.96 | 97.92 | 96.88 | 94.53 | 82.29 | 60.42 | 66.04 | 74.65 | 87.17 |
| xK-SR (Ours) | 1.63 (8.90) | 100.00 | 100.00 | 98.96 | 97.92 | 98.44 | 95.31 | 83.33 | 60.42 | 87.92 | 74.65 | 89.70 |
| ShadowKV | 1.68 (10.61) | 100.00 | 71.88 | 73.96 | 97.92 | 27.34 | 24.22 | 68.75 | 58.33 | 52.71 | 73.96 | 64.91 |
| xK-SR (Ours) | 1.68 (10.45) | 100.00 | 98.96 | 98.96 | 97.92 | 94.53 | 93.49 | 82.29 | 60.42 | 80.83 | 76.04 | 88.34 |
| ShadowKV | 1.71 (11.59) | 96.88 | 6.25 | 5.21 | 80.21 | 0.78 | 2.34 | 65.62 | 56.25 | 49.79 | 72.57 | 43.59 |
| xK-SR (Ours) | 1.70 (11.44) | 100.00 | 96.88 | 92.71 | 97.92 | 62.50 | 56.25 | 80.21 | 59.38 | 69.58 | 76.39 | 79.18 |
| ShadowKV* | 4.52 | 100.00 | 98.96 | 96.88 | 97.92 | 93.49 | 91.67 | 82.29 | 58.33 | 67.92 | 75.69 | 86.32 |
| xKV-SR (Ours) | 4.37 | 100.00 | 100.00 | 98.96 | 96.88 | 99.48 | 96.61 | 82.29 | 60.42 | 87.92 | 75.69 | 89.83 |
| ShadowKV* | 5.51 | 100.00 | 76.04 | 75.00 | 97.92 | 54.43 | 45.83 | 81.25 | 57.29 | 47.29 | 74.31 | 70.94 |
| xKV-SR (Ours) | 5.35 | 100.00 | 100.00 | 98.96 | 97.92 | 98.44 | 95.57 | 82.29 | 60.42 | 87.29 | 76.04 | 89.69 |

## D.2 RESULTS ON LONGBENCH

**KV-Cache Compression Results.** Table 7 presents the comprehensive evaluation of xKV against representative compression methods on the LongBench dataset, demonstrating consistent performance across diverse long-context tasks, including single-document QA, multi-document QA, summarization, few-shot learning, synthetic tasks, and code completion. Experiments were conducted on Llama-3.1-8B-Instruct and Qwen2.5-7B-Instruct-1M models.

MiniCache exhibits severe performance degradation, with accuracy dropping by 12.57% on Llama-3.1-8B-Instruct and a catastrophic 26.91% on Qwen2.5-7B-Instruct-1M compared to the baseline, reinforcing our earlier observation that cross-layer compression methods fail when token-wise cosine similarity assumptions are violated across different model architectures and task types.

At 8.03× compression, xKV achieves 42.27% average accuracy on Llama-3.1-8B-Instruct, closely matching PyramidKV and SnapKV. On Qwen2.5-7B-Instruct-1M, xKV reaches 40.37% accuracy, demonstrating competitive performance against PyramidKV and SnapKV, with a slight accuracy degradation.

These LongBench results validate xKV's robustness across heterogeneous task domains, confirming that our shared low-rank subspace approach effectively preserves critical information for diverse long-context reasoning scenarios while achieving aggressive compression rates comparable to leading token eviction methods.

**KV-Cache Compression with Selective Reconstruction Results.** In Table 8, we evaluate xK-SR and xKV-SR against Quest and ShadowKV baselines on the LongBench dataset using Llama-3.1-8B-Instruct. Quest achieves 42.63% accuracy through dynamic token loading with 8× GPU memory reduction via offloading, demonstrating minimal performance degradation while requiring host-device transfers.

At comparable compression ratios, xK-SR consistently outperforms ShadowKV across different settings. With 1.68× compression and 10.45× GPU memory reduction, xK-SR achieves 42.50% accuracy, surpassing ShadowKV by 1.99%. This improvement demonstrates the effectiveness of our cross-layer key compression approach over single-layer SVD methods.

Most notably, xKV-SR enables aggressive 5.35× compression while achieving 42.40% accuracy, outperforming ShadowKV* by 0.89%. These consistent improvements across both RULER and LongBench benchmarks validate that our cross-layer alignment approach effectively adapts to diverse evaluation frameworks, preserving critical information across heterogeneous long-context tasks ranging from retrieval and reasoning to code completion and summarization. Moreover, the significant gains observed on LongBench further corroborate the robustness and generality of our method beyond the RULER benchmark.

Table 7: KV-Cache Compression Results: Accuracy of different methods on LongBench. xKV consistently achieves a higher accuracy than the Full Attns at the same compression rate or even at a significantly higher compression rate.

| Method | Comp. | Single-doc QA | Multi-doc QA | Summarization | Few-shot | Synthetic | Code | Avg. |
|---|---|---|---|---|---|---|---|---|
| **Llama-3.1-8B-Instruct** | | | | | | | | |
| Full Attn | 1.00 | 44.23 | 44.72 | 28.52 | 25.88 | 53.44 | 62.41 | 43.20 |
| MiniCache | 1.30 | 22.01 | 26.79 | 20.51 | 25.05 | 52.29 | 37.11 | 30.63 |
| KIVI | 7.10 | 40.87 | 42.45 | 27.40 | 26.96 | 51.70 | 59.42 | 41.47 |
| StreamingLLM | 8.00 | 30.04 | 37.79 | 23.61 | 25.49 | 49.75 | 61.15 | 37.97 |
| PyramidKV | 8.00 | 42.92 | 43.99 | 25.73 | 27.62 | 53.02 | 61.54 | 42.47 |
| SnapKV | 8.00 | 43.17 | 44.13 | 26.09 | 27.75 | 53.27 | 62.56 | 42.83 |
| Single SVD | 8.40 | 30.34 | 23.93 | 20.26 | 27.41 | 44.75 | 52.63 | 33.22 |
| xKV (Ours) | 8.03 | 44.39 | 38.82 | 26.14 | 27.34 | 55.50 | 61.44 | 42.27 |
| **Qwen2.5-7B-Instruct-1M** | | | | | | | | |
| Full Attn | 1.00 | 40.52 | 49.30 | 26.67 | 37.07 | 54.00 | 45.11 | 42.11 |
| MiniCache | 1.30 | 11.29 | 15.93 | 6.87 | 28.51 | 5.38 | 23.21 | 15.20 |
| KIVI | 7.10 | 33.54 | 38.35 | 21.34 | 35.55 | 34.32 | 39.73 | 33.80 |
| StreamingLLM | 8.00 | 28.86 | 38.04 | 22.42 | 47.58 | 17.50 | 42.75 | 32.86 |
| PyramidKV | 8.00 | 39.48 | 48.31 | 23.32 | 44.13 | 54.00 | 43.33 | 42.09 |
| SnapKV | 8.00 | 40.21 | 48.32 | 24.93 | 43.73 | 54.00 | 44.19 | 42.56 |
| Single SVD | 8.40 | 39.13 | 43.32 | 24.04 | 32.00 | 36.25 | 38.42 | 35.53 |
| xKV (Ours) | 8.03 | 39.73 | 47.97 | 26.62 | 34.42 | 53.00 | 40.48 | 40.37 |

Table 8: KV-Cache Compression with Selective Reconstruction Results: Accuracy of different methods on the LongBench. Here, "Comp." indicates the total memory reduction, while the number in parentheses shows the effective GPU memory reduction considering KV-Cache offloading. ShadowKV* refers to a variant of ShadowKV that additionally compresses the value cache.

| Method | Comp. | Single-doc QA | Multi-doc QA | Summarization | Few-shot | Synthetic | Code | Avg. |
|---|---|---|---|---|---|---|---|---|
| **Llama-3.1-8B-Instruct** | | | | | | | | |
| Full Attn | 1.00 | 44.23 | 44.72 | 28.52 | 25.88 | 53.44 | 62.41 | 43.20 |
| Quest | 1.00 (8.00) | 43.18 | 44.40 | 28.20 | 26.57 | 52.88 | 60.55 | 42.63 |
| ShadowKV | 1.68 (10.61) | 37.98 | 44.11 | 25.26 | 24.43 | 53.35 | 57.92 | 40.51 |
| xK-SR | 1.68 (10.45) | 43.64 | 44.47 | 27.62 | 25.31 | 52.63 | 61.32 | 42.50 |
| ShadowKV | 1.64 (9.08) | 43.35 | 44.87 | 27.15 | 25.76 | 52.63 | 59.53 | 42.21 |
| xKV-SR | 1.63 (8.90) | 44.38 | 44.63 | 27.98 | 25.55 | 52.13 | 61.50 | 42.69 |
| ShadowKV* | 5.51 | 41.76 | 44.89 | 26.02 | 24.74 | 52.73 | 58.91 | 41.51 |
| xKV-SR | 5.35 | 44.58 | 45.20 | 27.76 | 25.32 | 52.63 | 58.94 | 42.40 |

## D.3 RESULTS ON REASONING-HEAVY TASK

To demonstrate the broad applicability of our method on reasoning-intensive benchmarks, we evaluated xKV on GSM8K and the BIG-Bench Hard (BBH) suite, where retention of all intermediate states is critical. As shown in Table 9, token eviction approaches suffer catastrophic declines, with accuracy plunging from approximately 78% to just over 10% on BBH and into the mid-50s on GSM8K at a 7× compression rate. Even the quantization baseline KIVI experiences significant degradation on BBH. In stark contrast, xKV preserves strong performance across both benchmarks, underscoring that our shared low-rank subspace compression achieves a consistently superior accuracy–compression trade-off, even under the most demanding reasoning conditions.

Table 9: Accuracy of different methods on GSM8K and BBH with Llama-3.1-8B-Instruct.

| Method | Comp. | GSM8K | BBH |
|---|---|---|---|
| Full Attn | 1.00 | 78.47 | 69.70 |
| PyramidKV | 7.00 | 54.66 | 10.89 |
| SnapKV | 7.00 | 59.06 | 10.59 |
| KIVI | 7.10 | 67.55 | 52.96 |
| xKV | 7.00 | 71.42 | 69.19 |

## D.4 INTEGRATE WITH QUANTIZATION

xKV can be combined with other cache management techniques. To illustrate this capability, we conducted preliminary experiments integrating xKV with Quantization. Specifically, we applied a simple round-to-nearest (RTN) quantization method to the compressed cache. With 4-bit quantization, the cache achieves a substantial 25.6× compression while maintaining model accuracy.

Table 10: Accuracy of xKV integrated with naive quantization on RULER benchmark.

| Method | Comp. | N-S1 | N-S2 | N-MK1 | N-MK2 | N-MQ | N-MV | QA-1 | QA-2 | VT | FWE | Avg. |
|--------|-------|------|------|-------|-------|------|------|------|------|------|------|------|
| **Llama-3.1-8B-Instruct** | | | | | | | | | | | | |
| Full Attn | 1.00 | 100.00 | 100.00 | 98.96 | 97.92 | 98.96 | 98.18 | 83.33 | 60.42 | 97.71 | 85.42 | 92.09 |
| xKV | 8.03 | 100.00 | 98.96 | 97.92 | 97.92 | 96.35 | 97.14 | 78.13 | 57.29 | 86.67 | 78.13 | 88.85 |
| xKV-4bit | 25.7 | 100.00 | 96.88 | 97.92 | 97.92 | 96.35 | 93.23 | 77.08 | 55.21 | 83.33 | 78.47 | 87.64 |
| xKV-3bit | 32.12 | 93.75 | 94.79 | 95.83 | 96.88 | 95.05 | 90.89 | 77.08 | 52.08 | 73.33 | 76.74 | 84.64 |

Table 10 presents the performance of xKV with naive quantization on the RULER benchmark, evaluated using Llama-3.1-8B-Instruct. We observe that xKV alone provides an 8× compression with minimal accuracy loss. Further applying 4-bit quantization yields a total compression of 25.6×, with only a slight drop in the average score from 88.85% to 87.64%. Even more aggressive 3-bit quantization achieves 32× compression, with a moderate decrease in performance (average 84.64%), demonstrating that xKV can be effectively combined with other cache reduction techniques without severely impacting accuracy.

### D.5 FLOPS & MEMORY COST

**Dense Reconstruction (no selection).** For a given type $\tau$, if one reconstructs all $L$ rows per layer at decode, the per-layer cost of $\mathbf{A}^\tau \mathbf{B}^\tau$ is

$$\text{FLOPs}_{\text{dense}}^{(\tau)} = L \cdot r_\tau \cdot d. \tag{3}$$

If both keys and values are reconstructed densely, costs add: $\text{FLOPs}_{\text{dense}} = \sum_\tau L r_\tau d$.

**Selective Reconstruction (per step).** With index sets $\mathcal{S}_{t,\ell,g}$ and $M_{t,\ell,g} = |\mathcal{S}_{t,\ell,g}|$, the per-step cost for a given $\tau$ becomes

$$\text{FLOPs}_{\text{sparse}}^{(\tau)} = \sum_{g=1}^{H_{\text{kv}}} M_{t,\ell,g}\, r_\tau\, d_h, \tag{4}$$

and $\text{FLOPs}_{\text{sparse}} = \sum_\tau \text{FLOPs}_{\text{sparse}}^{(\tau)}$ when compressing both types. When $M_{t,\ell,g} \ll L$, selective reconstruction is a small fraction of the dense cost. Computing $\mathcal{S}_{t,\ell,g}$ itself involves light matrix–vector operations and is independent of the cross-layer factors.

**Compressed-cache memory.** xKV stores, per group $k$, the shared token bases $\mathbf{A}_k^\tau \in \mathbb{R}^{L \times r_\tau}$ (one per type) and, per layer, the reconstructions $\mathbf{B}_\ell^\tau \in \mathbb{R}^{r_\tau \times d}$. Summed over all groups/layers, the total memory is

$$\sum_{\tau \in \{K^{\text{pre}}, V\}} \Big( \underbrace{\frac{N}{G} L\, r_\tau}_{\text{shared bases}} + \underbrace{N\, r_\tau\, d}_{\text{layer reconstructions}} \Big) + \underbrace{\frac{N}{c} d}_{\text{landmark (optional)}}, \tag{5}$$

compared to $2N L d$ for the full KV-Cache (keys and values). In Mode (A) (key-only compression), only the $K$ terms in (5) apply and values are fetched (or lightly quantized) from host memory. In Mode (B), both types are compressed and resident on GPU. When activating SR, we have to store the landmark $L_l$ of size $\frac{N}{c} d$ for computing the indices for selective reconstruction.

**How we compute the KV-Cache compression ratio.** Let $C$ denote the compression ratio achieved by the xKV cache (i.e., the ratio of the original KV-Cache size to the compressed size). The landmark requires storing $\frac{L}{8} \times d$ elements, which is exactly one-eighth the size of the full K-Cache. The outlier set is a constant and can be ignored when the context length is long. The numerator 2 in each formula represents the combined original size of keys and values; the denominator represents the post-compression storage of the KV-Cache plus the landmark set (and, in the total memory case, the full value cache).

For xK-SR (key-only compression, value offloading):

Effective GPU memory compression ratio:

$$R_{\text{xK-SR, GPU}} = \frac{2}{\frac{1}{C} + \frac{1}{8}} \tag{6}$$

Total memory compression ratio (counting values at original size):

$$R_{\text{xK-SR, total}} = \frac{2}{\frac{1}{C} + \frac{1}{8} + 1} \tag{7}$$

For `xKV-SR` (both keys and values compressed):

$$R_{\text{xKV-SR}} = \frac{2}{\frac{2}{C} + \frac{1}{8}} \tag{8}$$

## E    EXTENDING xKV ON MULTI-HEAD LATENT ATTENTION (MLA)

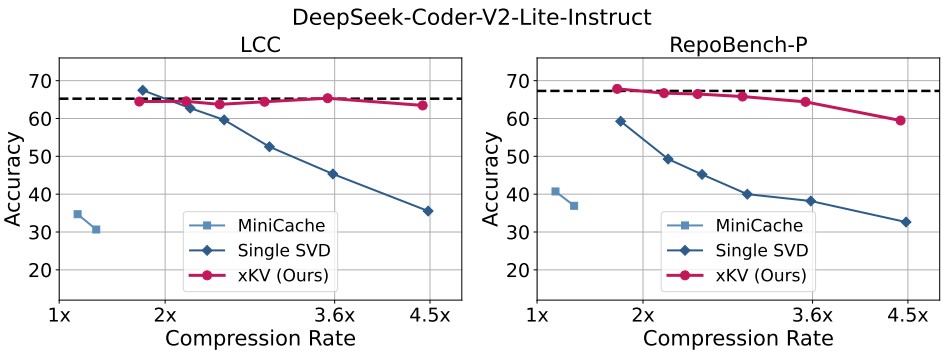

Figure 8: Evaluation results of different KV-Cache methods on DeepSeek-Coder-V2-Lite-Instruct model using RepoBench-P (Liu et al., 2023a) and LCC(Guo et al., 2023). The accuracy denotes the edit similarity (Svyatkovskiy et al., 2020), and the dotted line represents the baseline score with uncompressed KV-Cache.

To demonstrate the effectiveness of xKV on emerging attention variants, we evaluate xKV on DeepSeek-V2-Coder-Lite (Liu et al., 2024a), which employs the efficient Multi-head Latent Attention (MLA) architecture (Liu et al., 2024a). MLA is proposed to reduce the KV-Cache size per layer through low-rank projections. As shown in Figure 8, we can further compress the compact latent cache by exploiting the cross-layer redundancy by using our xKV. With a group size of 4, xKV achieves a 3× compression rate on RepoBench (Liu et al., 2023a) and 3.5× on LCC (Guo et al., 2023) without compromising accuracy. In contrast, other methods, such as MiniCache (Liu et al., 2024b) and Single SVD, fail to preserve accuracy on the MLA architecture even at substantially lower compression rates. These results underscore xKV's versatility and compatibility with emerging memory-efficient attention architectures (Liu et al., 2024a).

## F    BROADER CKA ANALYSIS

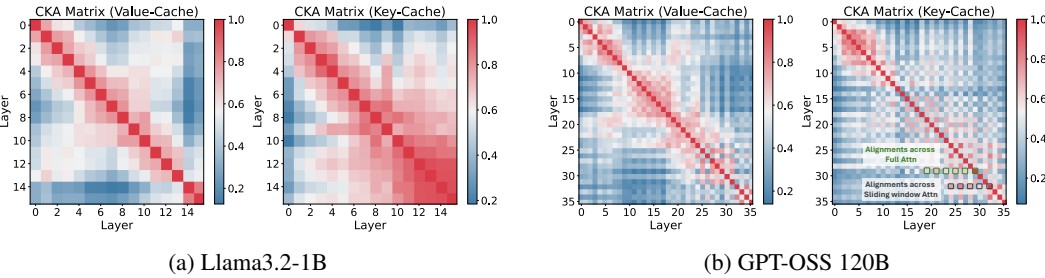

(a) Llama3.2-1B                                    (b) GPT-OSS 120B

Figure 9: Extended CKA analysis of different models. GPT-OSS is a hybrid architecture that interleaves window attention and full attention layers in a 1:1 ratio.

