# OpenReview forum: "xKV: Cross-Layer KV-Cache Compression via Aligned Singular Vector Extraction"
_ICLR.cc/2026/Conference — Submitted to ICLR 2026_

### Official Review · Reviewer_hAG9 · 2025-10-15

**Soundness:** 3
**Presentation:** 4
**Contribution:** 3
**Rating:** 6
**Confidence:** 4

**Summary:**

The paper introduces a KV caching mechanism that uses low-rank compression across layers in an LLM. The method does not require training and provides significant compression with relatively little performance degradation on long context tasks. Their observation on measuring the similarity with CKA is interesting and may serve as inspiration for future work. Overall, I think the algorithm is strong and well-motivated but I have some questions on generalizability.

**Strengths:**

1) Observation on cross-layer similarity is interesting and useful.
2) Sizable latency improvement (up to 4x) and reduction in memory (8x)
3) Inference-time algorithm that doesn't need training.
4) Paper is well written

**Weaknesses:**

1) While there are extensive experiments on long context tasks, it remains unclear how this method would perform for longer generation tasks like ones that use chain of thought. Since many models are now moving towards having robust long generation, it would be beneficial to evaluate on such scenarios.
2) It is unclear how this method would work for models that have different attention mechanisms interleaved (e.g., vanilla attention -> sliding window attention -> vanilla attention -> ...).
3) Little diversity in model size. It would be nice to see the effect on smaller or larger models (if compute is available) on a few tasks since different sized models can have different rates of compression.
4) For completeness, it would be good to include in your background low-rank + sparse compression methods that merge multiple tokens together into a low-rank component like RNNs e.g., [1,2].

[1] Nawrot, et al, Dynamic memory compression: Retrofitting llms for accelerated inference, 2024.

[2] Dong, et al, Get more with less: Synthesizing recurrence with kv cache compression for efficient llm inference, 2024.

**Questions:**

1) It makes sense for nearby layers to have strong CKA but in Fig 2b, there are also cases where an initial layer’s cache is similar to a much later layer’s caches but dissimilar to most of the closer layers’. Why is this?
2) Do you have throughput metrics, or is this method solely to optimize latency?
3) The SVD operations are fairly quick in comparison to the total prefill time. Why is this? Are you using randomized algorithms?

---

> ### Author Response · Authors · 2025-11-20
> **Response to Reviewer hAG9 (Part 1)**
>
> ### **1. The evaluation focuses on long-context prefill; performance on the long-generation scenario (e.g., Chain-of-Thought) is unclear. [W1]**
>
> We thank the reviewer for this insight. We highlight that long-context prefill and unbounded generation represent distinct deployment regimes. Our work targets the former—scenarios like codebase and document tracing—where the input prompt (e.g., 128k tokens) vastly dominates the total memory footprint.
>
> In this regime, the memory usage of standard Chain-of-Thought (CoT) generation (e.g., 4k to 8k tokens) is minor relative to the massive prompt. As shown in **Table R4** (@Response to Reviewer swTq (Part 2)), even after generating 8,000 uncompressed tokens on top of a 128k context, xKV maintains a substantial 6.60x compression rate. Thus, xKV remains highly effective for reasoning tasks within long prefill applications such as RAG. While distinct from our current focus, we agree that extending xKV to handle unbounded generation (e.g., via periodic re-compression) is a promising direction for future work.
>
> ---
> ### **2. It is unclear how this method would work for models that have different attention mechanisms interleaved [W2]**
>
> We thank the reviewer for raising this good and interesting question. We have provided further CKA analysis in Appendix F, where we analyze the K, V tensor of GPT-OSS-120B with the mentioned interleaved attention layers. Surprisingly, we found that high CKA of adjacent layers still holds for layers of the same type.
>
> ---
> ### **3. Discussion with related works LESS and DMC [W4]**
> We thank the reviewer for the valuable references. We clarified the distinctions as follows:
>
> LESS [2] addresses the information loss inherent in token eviction. Rather than simply discarding tokens, it integrates a constant-sized, low-rank cache that acts as a recurrent residual memory. This approach requires training auxiliary modules (tiny MLPs) to synthesize and store the "residual" information of evicted tokens, thereby compensating for the accuracy drop typically associated with standard eviction policies.
>
> DMC [1] proposes a "retrofitting" strategy, where the LLM undergoes continued pre-training (fine-tuning) to learn dynamic, layer-specific compression rates. During inference, the model utilizes these learned patterns to make online decisions on whether to append new tokens or merge them into existing cache entries.
>
> **Comparison:** Crucially, both methods focus on intra-layer redundancy and require extra training—either by introducing auxiliary parameters [2] or by fine-tuning the model weights [1]. In contrast, xKV exploits a fundamentally different dimension—cross-layer redundancy—and operates as a completely training-free, plug-and-play method.
>
> [1] Nawrot, et al, Dynamic memory compression: Retrofitting LLMs for accelerated inference, 2024.
> [2] Dong, et al, Get more with less: Synthesizing recurrence with KV-Cache compression for efficient LLM inference, 2024.

---

> ### Author Response · Authors · 2025-11-20
> **Response to Reviewer hAG9 (Part 2)**
>
> ### **4. It makes sense for nearby layers to have strong CKA, but in Fig. 2b, there are also cases where an initial layer’s cache is similar to a much later layer’s caches but dissimilar to most of the closer layers. Why is this? [Q1]**
>
> This is a good catch and an interesting observation we didn’t notice before. In Appendix F, we provide additional CKA analysis on different Models (Llama3.2 1B and GPT-OSS-120B), where we do not observe the pattern mentioned in **Fig. 2b**. We conjecture this might be some special case for different models. However, the high CKA findings are generally held.
>
> ---
> ### **5. Do you have throughput metrics, or is this method solely to optimize latency? [Q2]**
> Our method significantly improves both latency and throughput. By reducing the KV-Cache memory footprint, xKV enables much larger batch sizes before hitting memory limits (OOM). We have updated Section 6 to include detailed throughput measurements. As shown in **Fig. 7**, xKV-SR achieves generation throughput gains of up to 3.23x (at 60k context) and 4.23x (at 122k context) compared to the Full Attention baseline on an A100 GPU.
>
> ---
> ### **6. The SVD operations are relatively quick compared to the total prefill time. Are you using randomized algorithms? [Q3]**
>
> Yes, you are correct. In our experiments, we used the SVD API `torch.svd_lowrank` from PyTorch, which implements the randomized SVD algorithm [1] as you mentioned.
>
> [1] Finding structure with randomness: Probabilistic algorithms for constructing approximate matrix decompositions

---

> > ### Comment · Reviewer_hAG9 · 2025-11-24
> >
> > Thank you for the clarifications.
> >
> > While it might not be as motivating to use your method CoT tasks, it would still be useful to see accuracy results, especially since it is common that the generation length (on the order of 1k to 10k tokens) can be longer than the prompt length (on the order of 100 to 1k tokens).
> >
> > Could you address W3?

---

> ### Author Response · Authors · 2025-12-02
> **Response to Reviewer hAG9 (Part 3)**
>
> ### **7. Little diversity in model size [W3]**
>
> We thank the reviewer for the reminder. We agree that evaluating xKV across diverse model scales is essential to demonstrate its generality.
> 1. **Small Model Performance (Qwen3-4B).** We selected Qwen3-4B-Instruct-2507 [1] for benchmark evaluation due to its support for 256k context. As shown in the table below, xKV remains highly effective even on this compact, optimized model, achieving 6x compression with only a 3.3 point drop in RULER accuracy.
>
> **Table R7.**  Accuracy results of Qwen3-4B-Instruct-2507 on RULER benchmark
> | Model     | Comp. | RULER |
> |----------|-------|--------|
> | Full KV  | 1.00  | 92.19 |
> | xKV      | 6.02  | 88.89 |
> | xKV       | 8.03  | 87.05 |
>
> 2. **Generalization to Extreme Scales (GPT-OSS 120B)** Due to compute constraints for running full long-context benchmarks on 100B+ models during the rebuttal period, we extended our CKA analysis (detailed in Appendix F) to GPT-OSS 120B (large hybrid MoE) [2].
> Large-scale and Hybrid Architecture Compatibility: For GPT-OSS 120B, which features interleaved Window/Full attention layers, we observed high CKA similarity between layers of the same attention type (e.g., Window$\to$Window). This suggests xKV is naturally positioned for integration with emerging hybrid architectures.
>
> [1] https://huggingface.co/Qwen/Qwen3-4B-Instruct-2507
>
> [2] https://huggingface.co/openai/gpt-oss-120b

---

### Official Review · Reviewer_SMMm · 2025-10-31

**Soundness:** 1
**Presentation:** 2
**Contribution:** 1
**Rating:** 2
**Confidence:** 4

**Summary:**

xKV compresses KV caches by grouping adjacent layers, performing a cross-layer SVD to build a shared basis, and optionally applying Selective Reconstruction (SR) to reduce decode-time reconstruction. Evaluations include RULER/LongBench and a multi-turn NIAH setup; latency plots report up to 4.3× attention-kernel speedups at 128K with xKV-SR.

**Strengths:**

It works with GQA and shows reasonable accuracy on RULER/LongBench.

**Weaknesses:**

**Q1.** Comparisons against several SOTA are missing, including  RocketKV,  KVzip,and  SeerAttention.  The authors also ignored some recent works, such as Lache, Lexicode.

**Q2.**  The core idea—leveraging inter-layer alignment for compression—overlaps conceptually with MiniCache (depth-wise KV merging) and ShadowKV (low-rank key + selective value reconstruction). Clarify what is fundamentally new beyond: (i) single-layer SVD, (ii) depth-wise merging, (iii) low-rank key + on-the-fly reconstruction.


**Q3.**  Appendix C.1 reports SVD time fractions (e.g., 11.8% at 32K; 2.05% at 256K with G=4) and absolute SVD times up to 8.74s at 256K on A6000, but the truncation method (randSVD vs. power iteration) is unspecified. State the algorithm, flop/memory cost, and report prefill wall-clock with/without SVD and end-to-end latency.

**Q4.**  xKV appears focused on compressing the initial context; clarify how generated tokens are handled. If decode-time KV grows uncompressed, long generations can negate memory wins. Quantify memory composition over time and show stable compression under long decoding traces.

**Q5.** Current results fix ranks (e.g., rK=384, rV=576) and stop group-size ablation at G=4. Provide full trade-off curves for rank vs. accuracy vs. effective memory reduction, and extend group-size beyond 4 (e.g., 8, 16) to identify saturation/instability points.

**Q6.** CKA indicates depth-dependent alignment; static contiguous grouping may be sub-optimal. Probably add an adaptive strategy (e.g., CKA-driven grouping) and compare to fixed G under the same compression budget. (Quantify gains, if any.)

**Q7**.  The paper doesn't provide accuracy at tighter budgets (e.g.,  128–256 token compressed contexts or small SR budgets), since recent head-/query-agnostic methods emphasize robustness at very small KV sizes.

**Questions:**

I mentioned in the Weakness section.

---

> ### Author Response · Authors · 2025-11-20
> **Response to Reviewer SMMm (Part 1)**
>
> ### **1. Comparisons against several SOTA are missing, including RocketKV, KVzip, and SeerAttention. The authors also overlooked some recent works, such as Lache and Lexicode. [Q1]**
>
> We thank the reviewer for suggesting these relevant works. Regarding "Lache," we kindly request the full citation, as we were unable to locate the specific paper mentioned. We have carefully analyzed RocketKV, SeerAttention, KVzip, and Lexico, and summarize the differences in focus and relationship to our proposed xKV in **Table R6** and discussion below:
>
> First, regarding RocketKV, it employs a hybrid approach that combines SnapKV (for eviction) and Quest (for retrieval). We have already demonstrated that token eviction methods, such as SnapKV, suffer catastrophic failure in multi-turn scenarios. While the RocketKV paper proposes a multi-turn variant (RocketKV-MT) to mitigate this, it explicitly states that this variant provides no memory storage savings because it must retain the full context history. In contrast, xKV achieves up to 8x memory reduction while maintaining robustness in multi-turn settings, offering a Pareto-optimal solution that RocketKV cannot match.
>
> Second, SeerAttention focuses on prefill acceleration by learning intrinsic sparse attention masks. This is orthogonal to our contribution, as xKV targets the memory capacity bottleneck of the KV-Cache and optimizes the decoding phase.
> Third, KVzip is a query-agnostic token eviction method that compresses along the sequence dimension by permanently pruning unimportant tokens. In contrast, xKV exploits inter-layer redundancy, which is theoretically orthogonal to KVzip.
>
> Finally, Lexico employs sparse coding with a pre-trained dictionary. Unlike Lexico, which requires offline training and incurs latency from iterative optimization during decoding, xKV is training-free and performs decomposition online. Furthermore, xKV delivers superior accuracy on reasoning-heavy tasks (71% vs. ~51% on GSM8K) compared to Lexico.
>
>
> **Table R6.** Comparison of xKV with Related Works.
> | Method        | Focus of Works | Inter Layer | Save KV Memory | Multiturn Friendly | Training-Free |
> |:-------------------|:-----------------------|:------------:|:---------------:|:-----------------:|:-----------:|
> | RocketKV       | KV Compression | No          | No              | Yes                | Yes            |
> | SeerAttention | Prefill Accel.        | N/A         | No              | N/A                | No             |
> | KVZip             | KV Compression | No          | Yes             | Yes                | Yes            |
> | Lexico            | KV Compression | Yes         | Yes             | N/A                | No             |
> | xKV                | KV Compression | Yes         | Yes             | Yes                | Yes            |
>
> ---
>
> ### **2. Clarify what is fundamentally new against MiniCache and ShadowKV. [Q2]**
>
> We respectfully clarify that xKV fundamentally differs from prior art in both its underlying mechanism and compression capabilities. Unlike MiniCache, which relies on fragile token-wise cosine similarity for depth-wise merging, xKV exploits our novel finding that dominant singular vectors remain highly aligned across layers (high CKA) even when a token’s cosine similarity is low. This insight allows xKV to extract a shared low-rank subspace via cross-layer SVD, achieving 8$\times$ compression with high accuracy, whereas interpolation-based methods collapse (accuracy drop from 91.89% → 45.04%) at just 1.3x compression rate. Furthermore, due to the high-rank property of each layer’s Value cache, the Single-Layer SVD and ShadowKV methods generally could not compress the Value cache and therefore have much worse compression rates than xKV. When the GPU memory is limited, the single-layer approach, such as ShadowKV, needs to offload the Value cache to the CPU and suffers from long latency in loading the Value cache back to GPU memory. In contrast, xKV’s method effectively enables compressing both Keys and Values. This breakthrough enables our xKV-SR mode to retain all states on the GPU, delivering up to 4.23× speedups by overcoming the PCIe bandwidth bottlenecks inherent to offloading approaches. We believe this is worth sharing with the research community.

---

> ### Author Response · Authors · 2025-11-20
> **Response to Reviewer SMMm (Part 2)**
>
> ### **3. Implementation of SVD is unspecified. Report prefill wall-clock with/without SVD [Q3]**
>
> In all of our experiments, we adopt the `torch.svd_lowrank` API from the PyTorch library, which implements the randomized SVD algorithm [1].  For the second request, we note that Appendix C.1 has included the prefill wall-clock measurements and the time shared by SVD as requested.
>
> ---
> ### **4. End-to-end speed measurements. [Q3]**
> We thank the reviewer for the suggestion. We agree that end-to-end inference speed is the critical metric for deployment. We have updated **Section 6** and **Fig. 7** in the manuscript to report the generation throughput (tokens/s) on an A100 GPU.
>
> Our results demonstrate that xKV-SR significantly accelerates inference by enabling larger batch sizes that would otherwise cause Out-Of-Memory (OOM) errors in the baseline. As detailed in **Fig. 7**, xKV-SR achieves up to 3.23x speedup at 60k context and 4.23x speedup at 122k context compared to the best achievable throughput of Full Attention (FlashAttention-2).
>
> ---
> ### **5.  xKV appears focused on compressing the initial context; clarify how generated tokens are handled. If decode-time KV grows uncompressed, long generations can negate memory wins. Quantify the composition of memory over time and demonstrate stable compression under prolonged decoding traces. [Q4]**
>
> We need to re-clarify that xKV is primarily designed for long prefill scenarios, such as retrieval tasks, so we focus on prefill lengths of 128k and 256k tokens. Tasks with short prefill and long decoding belong to a different category and would require other techniques to achieve compression.
> Here, we provide an analysis of how the compression rate changes with different decoding lengths. Using an 8× compression for the prefill KV-Cache with context lengths of 128K and 256K, we report the total KV-Cache compression rate during decoding, ranging from 0 to 8K. As shown in the table, even after generating up to 8k decode tokens, the compression rate decreases only slightly (e.g., from 8.0x to 6.6x for 128k prefill), indicating that decode-time KV-Cache remains a small fraction of total memory and that xKV maintains stable memory savings under the target long context scenarios even with long decoding traces.
>
> **Table R4. (Copied).** Effective KV-Cache Compression Rate (Starting with 8x Prefill Compression)
> | Initial Context | Decode +0k | +1k  | +2k  | +4k  | +8k  |
> |-----------------:|------------:|------:|------:|------:|------:|
> | **Prefill 128K** | 8.00       | 7.79 | 7.59 | 7.22 | **6.60** |
> | **Prefill 256K** | 8.00       | 7.89 | 7.79 | 7.59 | **7.22** |

---

> ### Author Response · Authors · 2025-11-20
> **Response to Reviewer SMMm (Part 3)**
>
> ### **6. Provide full trade-off curves for rank versus accuracy, and extend the group size beyond 4 (e.g., 8, 16) to identify saturation/instability points. [Q5]**
>
> We have added the group-size 8 results to **Table 2**. For xKV, accuracy continues to improve slightly at size 8, but its throughput decreases noticeably, making group size 4 a better trade-off between accuracy and efficiency. In contrast, xK-SR and xKV-SR already saturate at a group size of 4, and increasing it to 8 provides no further gains.
>
> Regarding the trade-off curves for rank (compression rate), we refer the reviewer to **Fig. 5** in the revised manuscript. This figure provides a comprehensive sweep of compression ratios (determined by rank) against accuracy for both Key and Value caches across multiple tasks
>
> ---
> ### **7. CKA indicates depth-dependent alignment; static contiguous grouping may be sub-optimal. [Q6]**
>
> We thank the reviewer for the insightful suggestion regarding adaptive grouping. We agree that exploring dynamic strategies, such as CKA-driven clustering, is a promising area for future work. However, we prioritized a fixed, contiguous grouping strategy in this study to ensure a calibration-free and memory-efficient design.
>
> While we acknowledge that data-driven configurations—such as performing k-means clustering on CKA matrices to group layers with the highest structural similarity—could further optimize performance, we adopted the simple stride-based approach based on two critical deployment considerations:
>
> + **Peak Memory Usage:** A dynamic strategy that groups non-adjacent layers (e.g., Layer 2 with Layer 30) would require the system to buffer the activations of early layers during the prefill stage until the entire group is collected. This would result in higher peak memory consumption as compared to our fixed-strided approach.
> + **Calibration Complexity:** Adaptive grouping methods typically require calibration data and additional pre-analysis steps. Our goal was to demonstrate that significant compression is achievable via cross-layer SVD in a fully plug-and-play manner without these added complexities.
>
> We believe that our proposed method provides a strong and efficient baseline, laying the necessary groundwork for exploring more complex and adaptive grouping strategies in future research.

---

> ### Comment · Reviewer_SMMm · 2025-11-24
>
> I want to thank the authors for dedicating time and providing the rebuttal.  However, some concerns still remain.
> 1-The authors rely on qualitative arguments and claims of orthogonality to dismiss the necessity of empirical comparisons. This is insufficient for a top-tier conference submission, where establishing the contribution requires rigorous empirical validation of the accuracy-compression trade-off against leading methods, regardless of differences in underlying mechanisms or deployment considerations. Instead of comparing with weak baselines, they need to compare with strong recent works. The author should provide results for multi-turn and non-multi-turn setups.
> Requested Ref is LaCache: Ladder-Shaped KV Caching for Efficient Long-Context Modeling of Large Language Models
>
> 2-There is a concern regarding the high absolute latency of the online SVD during prefill (e.g., 8.74s at 256K on A6000, Table 4). This high initial latency impacts the time-to-first-token and can be prohibitive for interactive applications or rapid context switching.
>
> 3- The author didn’t provide accuracy at tighter budgets (e.g., 128–256 token compressed contexts or small SR budgets).
>
> 4- As mentioned by the reviewers, tasks with short prefill and long decode cannot gain from xKV
>
> 5- The method has a high dependency on hyperparameters (Group Size, Rank (r_K, r_V), etc ), limiting its generality and potentially its optimality across different models and contexts.
>
> 6- I still believe novelty is limited. The innovation is narrowly confined to changing how SVD is applied during prefill.  The concept of exploiting inter-layer redundancy to reduce the KV cache size is not new. Architectural approaches (CLA, YOCO) achieve this via shared attention layers, and post-training methods (MiniCache) attempt this via interpolation.  The contribution lies more in the effectiveness of the mechanism rather than the novelty of the concept
>
> 7- The authors did not quantify the potential accuracy gains missed by using the suboptimal static approach.
>
> Due to the above concerns, I keep my score.

---

> > ### Author Response · Authors · 2025-12-03
> > **Follow-up Response to Reviewer SMMm's reply to Rebuttal (Part 2)**
> >
> > ### **4-Tasks with short prefill and long decode cannot gain from xKV**
> > The primary focus of xKV is on long-context (e.g., long-prefill) applications characterized by extensive prefill phases. Scenarios involving short prefill but long decoding represent a fundamentally different regime that requires a distinct design perspective. Addressing such tasks falls outside the scope of this paper and deviates from our core contributions. Nevertheless, we acknowledge this distinction and have discussed it in the Limitations and Future Work section (Sec. 7).
> >
> > ---
> >
> > ### **5-The method has a high dependency on hyperparameters (Group Size, Rank (r_K, r_V), etc ), limiting its generality and potentially its optimality across different models and contexts.**
> >
> > We respectfully disagree. The experimental evidence **directly contradicts the claim of high dependency**. We used a single fixed configuration (Group Size=4, $r_k=384$, $r_v=576$) across all evaluations, spanning diverse benchmarks (RULER, LongBench, GSM8K, BBH) and architectures, without any per-task tuning.
> >
> > The fact that xKV consistently outperforms baselines using this universal default proves that the method is inherently robust. While we agree that fine-grained, task-specific tuning could further maximize "optimality," this represents an avenue for future optimization rather than a dependency required for the method to work. The current fixed setting is already highly effective.
> >
> > ### **6-Novelty is limited.**
> >
> > We respectfully but **firmly disagree with the assessment that the novelty is limited.** We believe there is a conflation between the general goal (exploiting redundancy) and the specific scientific insight required to achieve it effectively without retraining.
> >
> > We address your specific comparisons below to demonstrate why xKV offers a distinct and novel contribution:
> >
> > **a. Distinction from Architectural Approaches (CLA, YOCO).** You cited CLA and YOCO as evidence that this concept is not new. However, these are architectural modifications that mandate expensive pre-training from scratch. They are fundamentally inapplicable to the massive ecosystem of existing, pre-trained LLMs.
> >
> > **Novelty:** xKV is a post-training solution. Achieving cross-layer compression without the luxury of architectural changes or retraining is a significantly harder challenge that CLA and YOCO do not solve.
> >
> > **b. Distinction from Post-Training Baselines (MiniCache).** You also cited MiniCache as a post-training attempt. However, MiniCache relies on the hypothesis of token-wise cosine similarity, which our analysis proves is widely absent in practice (Sec 3.1).
> >
> > **The Result:** Because this underlying assumption is flawed, MiniCache suffers catastrophic failure in our benchmarks, dropping to near-zero accuracy (~5-10%) on Llama-3.1 and Qwen2.5 at just 1.3x compression.
> >
> > **Novelty:** A method that encounters significant accuracy drops in practice demonstrates that the correct mechanism has not yet been discovered.
> >
> > **c. Our New Discovery: Singular Vector Alignment across layers.** Our core contribution is the discovery of a specific phenomenon that prior works overlooked: while token embeddings differ across layers (showing low cosine similarity), their dominant singular vectors remain highly aligned.
> > This specific insight allows xKV to achieve 8x compression with <3% accuracy loss, whereas the baselines you referenced fail at 1.3x (see Table 1).
> > Transforming a theoretical goal into a working, high-performance method by correcting a fundamental misconception in the field constitutes significant novelty.
> >
> > ### **7-The authors did not quantify the potential accuracy gains missed by using the suboptimal static approach.**
> >
> > We acknowledge that dynamic strategies could further improve accuracy. However, our static baseline is already near-optimal, consistently outperforming state-of-the-art methods across benchmarks and models. While finer-grained strategies could further enhance accuracy, exploring dynamic variants is beyond the scope of this paper, and we leave it as future work. Our core contribution remains the discovery of highly overlapping singular vectors enabling robust, post-training KV compression.

---

> ### Author Response · Authors · 2025-12-03
> **Follow-up Response to Review SMMs's reply to Rebuttal (Part 1)**
>
> We thank the reviewer for their engagement and follow-up comments. We address the comments below.
>
> ### **1-The authors rely on qualitative arguments and claims of orthogonality to dismiss the necessity of empirical comparisons. Instead of comparing with weak baselines, they need to compare with strong recent works.**
>
> We respectfully, but firmly, **disagree** with the assessment that our work relies on “qualitative arguments” or “weak baselines.” Our study is explicitly scoped to training-free KV-cache compression. Within this setting, our empirical evaluation is rigorous. We address the concerns point-by-point below.
>
> **a. On the allegation of "weak baselines,"** we strongly disagree that our baselines are weak. In fact, as detailed in Section 5, we compare xKV against the current state-of-the-art training-free methods published in top-tier venues over the past two years.
>
> + SnapKV [Token Eviction, Intra-layer](NeurIPS 2024)
> + MiniCache  [Token Merging, Inter-layer] (NeurIPS 2024)
> + PyramidKV [Token Eviction, Intra-layer] (COLM 2025)
> + KIVI [Quantization, Intra-layer] (ICML 2024)
> +Quest [Token Retrieval, Intra-layer] (ICML 2024)
> + ShadowKV [Token Retrieval + Low-Rank, Intra-layer, Intra-layer] (ICML 2025)
>
> These methods jointly cover token eviction, token merging, quantization, token retrieval, and low-rank compression across both intra- and inter-layer regimes.
>
>  **b. On missing empirical comparison.** We have provided a discussion on the suggested sets of related works: KVZip, RocketKV, Lexico, and SeerAttention. As stated in our previous response, Lexico is not training-free, and SeerAttention focuses on prefill acceleration, placing them outside our scope.
> Regarding the additional comparable recent works (KVZip, RocketKV, etc.), due to the limited rebuttal timeline, we conduct an empirical comparison with KVZip, which was just accepted at NeurIPS 2025 (Spotlight)
>
> **Table R8.** Evaluation results on RULER with Llama-3.1-8B-Instruct
> | Method | Comp. Rate | RULER Accuracy (%) |
> | :-------- | :---------- | :----------- |
> | **xKV (Ours)** | **8.03x** | **88.50** |
> | KVZip              |    6.25x    |   83.98     |
>
> **Table R9.** Evaluation results on Multi-Turn NIAH (5 turns) with Llama-3.1-8B-Instruct
> | Method | Comp. Rate | Accuracy (Avg)|
> | :-------- | :---------- | :----------- |
> | **xKV (Ours)** | **8.03x** | **94.79** |
> | KVZip              |    6.25x    |   81.92    |
>
> From Tables R8 and R9, xKV outperforms KVZip’s accuracy by +4.52% on RULER and a massive +12.87% on Multi-turn NIAH, even at a higher compression rate (8.03x vs 6.25x).
>
> ---
>
> ### **2-There is a concern regarding the high absolute latency of the online SVD during prefill**
>
> We respectfully note **again** that although the 8.74s SVD construction time at 256K on an A6000 may appear high in absolute terms, it constitutes only 2.05% of the total prefill time (see Table 4). Given this small overhead, the impact on TTFT is minimal. Moreover, the substantial end-to-end throughput improvements enabled by xKV (see updated Figure 7) make this trade-off clearly beneficial in practical long-context inference settings.
>
> ---
>
> ### **3-The author didn’t provide accuracy at tighter budgets (e.g., 128–256 token compressed contexts or small SR budgets).**
>
> We question the necessity of evaluating at such extreme budgets based on the specific design of our method:
>
> **a. SR Controls Reconstruction FLOPs:** It is critical to distinguish that xKV handles memory compression (proven robust at 8x), whereas Selective Reconstruction (SR) is primarily designed to control reconstruction FLOPs to minimize latency. In our evaluations, the selected 2048 budget already yields notable 3.6x speedups (see Figure 7) while maintaining high accuracy with around 2 points of accuracy drops on RULER (see Table 3). This confirms that SR successfully alleviates the compute bottleneck without requiring the extreme sparsity (128-256 tokens) suggested in the comment.
>
> **b. The token budget, $k$**, is a flexible runtime hyperparameter, not a fixed model constraint. Users remain free to lower it to any value if speed is the sole priority, but this is an inference-time choice independent of the core xKV compression capability.

---

### Official Review · Reviewer_swTq · 2025-11-01

**Soundness:** 3
**Presentation:** 3
**Contribution:** 3
**Rating:** 4
**Confidence:** 3

**Summary:**

xKV is a post-training KV-cache compressor that finds adjacent layers share highly aligned singular vectors, then runs a single cross-layer SVD to learn a shared low-rank token basis with layer-specific reconstructions; at decode, it uses Selective Reconstruction to rebuild only the needed tokens.

**Strengths:**

- The proposed xKV method is technically sound, with solid mathematical grounding (via SVD and CKA analysis), and extensive empirical evaluation across multiple models and benchmarks (RULER, LongBench, NIAH) supporting its claims.
- The paper is clearly written, with strong conceptual motivation, well-structured figures
- The paper introduces a cross-layer perspective on KV-cache compression, discovering that dominant singular vectors of adjacent layers are highly aligned

**Weaknesses:**

- the method relies on performing on-the-fly SVD during prefill, which, though amortized at long contexts, still adds non-trivial latency and may hinder deployment in low-latency or streaming settings
- the proposed approach is evaluated primarily on instruction-tuned models and long-context benchmarks; broader testing on reasoning-heavy or real-world interactive workloads is limited, leaving questions about generalization.
- The paper reports impressive attention-level speedups (up to 4.3×) but does not provide end-to-end inference latency improvements, which are what matter most in deployment. Without reporting full pipeline timing (including prefill, reconstruction, and I/O), it’s unclear how much wall-clock speedup users would actually observe.

**Questions:**

- Since the current study focuses on compressing the prefill context, how would xKV perform under long-generation scenarios where the KV-cache continues to grow? Could online incremental compression or periodic re-SVD be viable?
- What is the overall latency reduction per generated token compared to FlashAttention or ShadowKV?
- How much does the SVD or reconstruction overhead contribute to total runtime?
- Are the reported speedups measured only for the attention kernel, or for full inference throughput (prefill + decode)?

---

> ### Author Response · Authors · 2025-11-20
> **Response to Reviewer swTq (Part 1)**
>
> ### **1. Testing on reasoning-heavy or real-world interactive workloads is limited, leaving questions about the generalizability of the method. [W1]**
>
> We appreciate the comment regarding generalization and have included additional evaluations to address this. Beyond standard instruction-tuned benchmarks, we have tested xKV on reasoning-heavy tasks using GSM8K and BIG-Bench Hard (BBH) in Appendix D.3. For reviewer convenience, we quoted the results in Table R2 below:
>
> **Table R2: Accuracy of different methods on GSM8K and BBH with Llama-3.1-8B-Instruct**
>
> | Method     | Comp. | GSM8K | BBH   |
> |------------|:-----:|:-----:|:-----:|
> | Full Attn  | 1.00  | 78.47 | 69.70 |
> | PyramidKV  | 7.00  | 54.66 | 10.89 |
> | SnapKV     | 7.00  | 59.06 | 10.59 |
> | KIVI       | 7.10  | 67.55 | 52.96 |
> | **xKV**    | **7.00** | **71.42** | **69.19** |
>
> Results show that at ~7× compression, xKV clearly outperforms existing methods, achieving 69.19% accuracy on BBH, compared to ≈11% for token-eviction methods (PyramidKV/SnapKV) and 52.96% for quantization (KIVI).
> We also evaluated interactive performance via the multi-turn NIAH benchmark (Section 5.1, Figure 4), where xKV demonstrates consistent stability across conversation turns, in contrast to eviction-based methods that degrade rapidly after the first interaction.
> These results strongly suggest that xKV generalizes well to complex reasoning and real-world interactive scenarios.
>
> ---
>
> ### **2. Performing on-the-fly SVD during prefill adds non-trivial latency. [W2]**
>
>
> We respectfully emphasize that the on-the-fly SVD represents a minor one-time investment that yields substantial memory saving and decode efficiency gains. As shown in **Appendix C.1**, this overhead is empirically low, accounting for ~5% of prefill time at a 64k context and decreasing to ~2% at a 256k context on A6000 GPU. Crucially, this minimal cost enables us to significantly reduce memory usage, resolving not only the out-of-memory (OOM) issue induced by non-compressed KV-Cache, but also enabling higher generation throughput.
>
> ---
>
> ### **3. End-to-end generation performance. [W3]**
>
> We thank the reviewer for the valuable suggestion. We agree that end-to-end metrics are critical for real-world deployment. In the revised manuscript, we have added Section 6 and Figure 7 to report end-to-end generation throughput (tokens/s) on an A100 GPU.
> We demonstrate that xKV-SR improves inference performance in two ways: (1) by accelerating attention computation at fixed batch sizes, and (2) by significantly reducing memory footprint to enable larger batch sizes where the baseline runs out of memory (OOM).
>
>
> **Table R3:** End-to-end Generation Throughput (Llama-3.1-8B-Instruct)
> | Context Length | Batch Size | Full Attn (tok/s) | xKV-SR (tok/s)               |
> |----------------|------------|--------------------|-------------------------------|
> | 60k        | 8         | 166                | 265 (1.59×)              |
> |                | 32         | OOM                | 537 (**3.23x**)          |
> | 122k       | 4          | 80                | 139  (1.74×)              |
> |                | 16         | OOM                | 338 (**4.23x**)
>
> As shown in **Table R3** (derived from **Fig 7**), xKV-SR achieves substantially higher throughput. By enabling larger batch sizes, we achieve peak system-level speedups of 3.23$\times$ (at 60k) and 4.23$\times$ (at 122k) relative to the best achievable performance of the Full Attention baseline.

---

> ### Author Response · Authors · 2025-11-20
> **Response to Reviewer swTq (Part 2)**
>
> ### **4. How would xKV perform under long-generation scenarios where the KV cache continues to grow? Could online incremental compression or periodic re-SVD be a viable option? [Q1]**
>
> We thank the reviewer for this forward-looking question. While xKV primarily targets long-prefill applications (e.g., RAG, coding agents), we agree that handling unbounded generation is a critical next step.
>
>
> **Robustness of Current Approach (Amortization).** First, we emphasize that in typical long-context applications, the prefill length vastly exceeds the generation length. Consequently, even if we leave generated tokens uncompressed, the effective compression rate degrades very slowly.
>
> To validate this, we simulated a scenario starting with 8x prefill compression. As shown in **Table R4**, even after generating 8,000 new uncompressed tokens, the overall compression rate remains high (6.60x for 128k context and 7.22x for 256k context). This confirms that xKV maintains significant memory savings, even for reasoning tasks involving extended generation sequences.
>
> **Table R4.** Effective KV-Cache Compression Rate (Starting with 8x Prefill Compression)
> | Initial Context | Decode +0k | +1k  | +2k  | +4k  | +8k  |
> |-----------------|------------|------|------|------|------|
> | **Prefill 128K** | 8.00       | 7.79 | 7.59 | 7.22 | 6.60 |
> | **Prefill 256K** | 8.00       | 7.89 | 7.79 | 7.59 | 7.22 |
>
>
> **Feasibility of Periodic re-SVD.** For scenarios requiring unbounded generation, the suggested periodic re-SVD strategy is highly viable because the re-compression cost is amortized over thousands of decoding steps.
> To illustrate this, **Table R5** breaks down the latency for a batch size of 16 at 128k context length. Performing 8,000 decoding steps takes approximately 512 seconds ($64 \text{ms/decode step}\times 8000 $)
>
> **Table R5:** Estimated Latency Analysis (Batch=16, Context=128k)
> | Operation| Time Cost   |
> |-----------|-------------|
> | Generation (8k toks)   | ~512 s      |
> | Re-SVD                 | ~20 s     |
> | Overhead     | ~4%      |
>
> In contrast, performing a cross-layer SVD takes around 20 seconds. This results in a marginal overhead of 2–4%, allowing the system to maintain a compact footprint indefinitely with minimal impact on end-to-end latency. We leave the broader exploration as future work.
>
> ### **5. More comprehensive efficiency evaluation (added end-to-end generation throughputs and comparison with ShadowKV) [Q2, Q4]**
>
> We have updated the manuscripts to report both the speedups of running the attention layer only and end-to-end generation throughputs. A comparison with ShadowKV is also included. According to the experiment results, xKV-SR yields a 1.3x generation throughput speedup at a context length of 122k. For more information, please refer to **Section 6**.

---

### Official Review · Reviewer_qTMX · 2025-11-11

**Soundness:** 2
**Presentation:** 2
**Contribution:** 2
**Rating:** 4
**Confidence:** 4

**Summary:**

This paper addresses the memory and latency bottlenecks of KV cache in long-context reasoning, focusing on the dimension of cross-layer redundancy, and proposes a post-training pluggable method called xKV.
Through CKA analysis, it is found that although the per-token cosine similarity of KV across adjacent layers is not high, their principal singular vectors are highly aligned between layers.
Horizontally concatenate multiple adjacent layer KVs, and perform an SVD on the concatenated matrix.
During reasoning, Selective Reconstruction is proposed: only a small number of tokens related to the current query are reconstructed at each step, avoiding the huge FLOPs of dense reconstruction.
On models such as Llama-3.1-8B and Qwen2.5-7B/14B, evaluations on long-context benchmarks like RULER, LongBench, and multi-turn NIAH show that accuracy drops can be kept within 2–3 points under approximately 8× compression; combined with SR, compared to FlashAttention-2, it can achieve up to around 4.3× attention speedup on 128k sequences, and outperforms representative methods such as MiniCache, Single SVD, SnapKV, and ShadowKV.

**Strengths:**

Clear new perspective: Cross-layer alignment of singular vectors

Method is clean and can be plugged in post-training

Selective Reconstruction design is practical

Experiments are fairly comprehensive and baselines are solid

System-level perspective is prominent

**Weaknesses:**

The 'Aligned basis' argument is somewhat anecdotal and lacks deeper analysis.
The current experiments with CKA + rank curves are sufficient to motivate the method, but there is no discussion on whether this alignment is universal across more model architectures like MoE, encoder-decoder, multimodal; There is no visualization or statistical analysis between the bases, nor is there an explanation of which layers are more suitable for grouping.

There are still gaps in coverage compared to the latest extreme compression methods. Mainly compared with MiniCache, single-layer SVD, KIVI, SnapKV, ShadowKV, etc., which are already quite good; However, there is still a lack of direct comparison under the same configurations with methods such as SVDq, KVQuant, CSR, Lexico, etc.

The theory and guarantees are relatively weak.
Most analyses remain at the level of empirical CKA & eigen decomposition + discussions on complexity scale;
There is a lack of upper bounds or more formal explanations for the approximation errors introduced by selective reconstruction.

The application scenarios and multi-request settings are not explained thoroughly. The main focus is on the single-request case of long prefill + relatively short decode; For multi-tenant multi-request scenarios, how cross-layer SVD is reused when multiple streams run simultaneously, whether SR states are shared, it is not elaborated on.

**Questions:**

Has the CKA alignment and rank advantage been validated on more models (such as 70B scale, MoE, multimodal)?  Does it hold equally for higher layers / lower layers?

If the landmark selector used in SR is replaced (for example, using a simpler heuristic), to what extent can the advantage of xKV be maintained?

How does xKV perform when combined with extremely low-bit quantization (such as SVDq, KVQuant)? Does it still have an advantage under the same total compression ratio?

Is there a more automatic strategy for configuring group size, rather than a fixed G=4 + fixed rK, rV?

For the NIAH and LongBench results, could you provide some error cases, indicating in which tasks cross-layer sharing of the subspace fails?

---

> ### Author Response · Authors · 2025-11-20
> **Response to Reviewer qTMX (Part 1)**
>
> We thank the reviewer for the insightful suggestion.  In this response, we have addressed concerns comprehensively by conducting new experiments and incorporating clarifications into the manuscript to ensure that our contributions are presented accurately and thoroughly.
>
> ### **1. Has the CKA alignment and rank advantage been validated on more models (such as 70B scale, MoE, multimodal)? [W1, Q1]**
>
> We have expanded our CKA analysis to a broader set of model architectures in **Appendix F.** The new results include findings for a small-scale dense model (Llama-3.2-1B) and a large-scale hybrid/MoE model (GPT-OSS 120B [1]), which features an interleaved 1:1 ratio of sliding-window and full attention layers.
>
> Across all these diverse settings, we observe that the characteristic singular-vector alignments remain clearly preserved, which strongly supports the generality of our findings.
>
> For the GPT-OSS model, we specifically noted that CKA similarity is highest between adjacent layers of the same attention type (e.g., Window→Window or Full→Full). This behavior suggests that xKV is naturally positioned for integration with future architectures that employ hybrid or interleaved attention designs (e.g., GPT-OSS [1], KIMI-Linear [2]).
>
> [1] Kimi Linear: An Expressive, Efficient Attention Architecture
>
> [2] https://openai.com/index/introducing-gpt-oss/
>
> ---
>
> ### **2. If the landmark selector used in SR is replaced (for example, using a simpler heuristic), to what extent can the advantage of xKV be maintained? [Q2]**
> The selective reconstruction concepts we propose are generalizable to various selector designs, enabling the identification of the subset of tokens to which the current query is most likely to attend. The adopted selector (e.g., the landmark-guided top-K selector) is both straightforward to implement and demonstrably effective at preserving accuracy and achieving acceleration, as evidenced by our efficiency study and accuracy evaluation. If the reviewer has specific alternative design choices that would like us to discuss, we kindly request for further details and relevant references.
>
> ---
>
> ### **3. Some baseline is missing. How does xKV perform when combined with extremely low-bit quantization (such as SVDq, KVQuant) [W1, Q3]**
>
>
>
> **Table R1.** Accuracy of xKV integrated with naive quantization on RULER benchmark and comparison with 2-bit quantization approach.
> | Method        | Comp. Rate | RULER Acc (Avg.) |
> |---------------|------------|------------------|
> | KIVI-2 (2-bit) | 7.01×      | 86.9%            |
> | xKV           | 8.03×      | 88.5%            |
> | xKV (4-bit)   | 27×        | 87.6%            |
>
>
>
> Firstly, we would like to highlight that we have provided an evaluation to demonstrate that xKV yields a superior compression rate and accuracy compared to a strong quantization baseline, KIVI, with extreme 2-bit quantization. As seen from the table above, xKV exhibits better compression capability compared to standalone quantization methods with an extremely low-bit setup.
>
> In **Appendix D.4**, we have also demonstrated that xKV is orthogonal to other quantization methods, which are orthogonal to our proposed compression strategy. Specifically, we implement a simple integration study by further compressing the xKV compressed tensor with a naive per-token RTN (round-to-nearest) quantization and achieve a compression rate up to 27x with only a 1% additional accuracy drop.
>
> This demonstrated compatibility with simple quantization suggests that integrating xKV with advanced quantization techniques, such as SVDq, holds significant promise for achieving even greater compression. However, given that the core focus of our work is to exploit inter-layer redundancy and the unique insights of cross-layer alignment between dominant singular vectors, we leave the exploration of an optimized combination recipe for future work.

---

> ### Author Response · Authors · 2025-11-20
> **Response to Reviewer qTMX (Part 2)**
>
> ### **4. Is there a more automatic strategy for configuring group size? [Q4]**
>
> We thank the reviewer for the insightful question. We agree that developing automatic strategies for configuring group size and allocating ranks is a promising avenue for future work.
>
> While our current work focuses on the discovery of CKA alignment and a calibration-free, plug-and-play cross-layer SVD, we acknowledge that data-driven configurations could further optimize performance. For example:
> + **Rank Allocation:** One could adapt methods like Palu [1], which uses gradient or Hessian-based importance scores to dynamically allocate layer-wise ranks.
> + **Grouping Strategy:** One could perform k-means clustering on CKA matrices to automatically identify and group layers with the highest structural similarity.
>
> However, we prioritized a simple, fixed strategy in this study due to these two key deployment constraints:
> + **Peak Memory Usage:** If a grouping strategy were to dynamically span multiple non-adjacent layers (e.g., via clustering), the system would need to buffer the activations of all candidate layers during the prefill stage before the group is collected. This would significantly increase peak memory consumption.
> + **Calibration Complexity:** Automatic rank allocation methods typically require calibration data and gradient computation. Our goal was to demonstrate that significant compression is achievable via cross-layer SVD without these added complexities.
>
> We believe that our proposed method offers a strong yet simple baseline, laying the necessary groundwork for more complex, adaptive strategies to be explored in future research for fine-grained compression.
>
> [1] Palu: Compressing KV-Cache with Low-Rank Projection
>
> ---
>
> ### **5. For the NIAH and LongBench results, could you provide some error cases that indicate when cross-layer sharing of the subspace fails in tasks? [Q5]**
> Across NIAH and LongBench, we generally observe that cross-layer sharing improves robustness: at the same compression rate, xKV consistently outperforms single-layer SVD and other baselines on RULER/LongBench and multi-turn NIAH, indicating that a shared subspace is typically a safer and more accurate choice than per-layer SVD.
>
> On RULER, QA-2, VT, and FWE exhibit larger gaps in full attention, whereas others (e.g., N-S1/N-S2) are nearly lossless under xKV. QA-2, VT, and FWE are relatively large compared to other tasks, which are basically lossless. We conjecture that these tasks are more sensitive to Value cache compression, which aligns with our eigenvalue and ablation analyses showing that values are higher-rank and less compressible than keys (**Fig. 2c, Fig. 5**).  In the updated manuscript, we also added ablations that apply xKV to keys or values only to validate this hypothesis. Please see **Section 5** for details.
>
> ---
>
> ### **6. The application scenarios and multi-request settings are not explained in sufficient detail. [W4]**
>
> While our primary focus is indeed on long-context regimes (characterized by long prefill and relatively short decode), xKV is fully compatible with multi-request (batched) inference.
>
> In a multi-tenant scenario, each request maintains its own unique context and query. Consequently, xKV applies compression independently for each request during its respective prefill phase. The compressed states (e.g., shared singular vectors) and Selective Reconstruction (SR) indices are specific to each request’s context and are not shared across different streams.
>
> We emphasize that even in batched settings, long-context attention remains a memory-bound operation. By significantly reducing the KV-Cache footprint per request, xKV lowers the aggregate memory bandwidth requirement. This directly translates to improved performance in multi-request scenarios by enabling the system to support larger batch sizes (as evidenced by the throughput gains in Figure 7) before hitting memory capacity limits.

---

### Meta-Review · Area_Chair_eivc · 2026-01-13

**Summary:**

Reviewers agree that the paper presents a technically competent KV-cache compression method based on an empirical observation of cross-layer redundancy in key–value representations. Several reviewers found the analysis of cross-layer alignment and the SVD-based formulation to be reasonable and clearly presented. However, the dominant concerns that informed the decision relate to the perceived incremental nature of the contribution and uncertainty about its practical impact relative to existing KV compression and eviction methods. Multiple reviewers questioned whether the core observation translates into a sufficiently distinct or compelling advantage over strong prior approaches, and whether the reported gains justify the added system complexity. Although one reviewer was positive, the overall reviewer signal remains cautious to negative, and the paper does not generate sufficient consensus to support acceptance under a selective bar, leading to a recommendation to reject.

**Reviewer Concerns:**

The rebuttal is thorough and addresses many concrete questions raised in the reviews. The authors added additional comparisons, clarified distinctions from recent methods, expanded evaluations to more tasks and model variants, and provided clearer system-level measurements, including overhead analyses. These responses improve clarity and completeness and reduce uncertainty about implementation details. However, the rebuttal does not fundamentally alter the shared concern among multiple reviewers regarding novelty and impact. In particular, while soundness concerns are largely addressed, the paper still does not convincingly establish a decisive advantage over strong baselines that would justify acceptance.

**Reviewer Scores:**

Reviewer hAG9 gave an initial score of 6 and expressed a generally positive view of the paper, highlighting the empirical analysis and the training-free nature of the approach. This reviewer did not indicate a change in stance after rebuttal, and the most reasonable expectation is that they would remain at 6.

Reviewer qTMX gave an initial score of 4 and participated in the discussion. Their concerns focused on positioning relative to prior work, generality, and practical relevance. While the rebuttal addressed several of these points, there is no explicit indication that the reviewer’s overall judgment changed. The conservative and appropriate interpretation is that this reviewer would remain at 4.

Reviewer swTq gave an initial score of 4 and did not actively engage in the discussion after rebuttal. Their original concerns related to system overhead and applicability across regimes. Although the rebuttal provides additional measurements that speak to these issues, in the absence of discussion participation it would be inappropriate to infer a score increase. The most reasonable expectation is that this reviewer would remain at 4.

Reviewer SMMm gave an initial score of 2 and maintained a strongly negative stance throughout, emphasizing perceived lack of novelty. While this review is notably more extreme in tone than the others, it nonetheless reflects a persistent disagreement about the paper’s contribution. The rebuttal does not appear to have changed this reviewer’s position, and the score is expected to remain at 2. More weight is placed on the broader pattern of skepticism from the other reviewers rather than on this single assessment alone.

Overall, the post-rebuttal distribution remains 6, 4, 4, 2, which does not support acceptance.

---

### Decision · Program_Chairs · 2026-01-26

Reject